# Neural Stochastic Control

**Jingdong Zhang**[1]     **Qunxi Zhu**[2,*]     **Wei Lin**[1,2,3,4,5*]

[1] School of Mathematical Sciences, SCMS, SCAM, and CCSB, Fudan University
[2] Research Institute of Intelligent Complex Systems
and MOE Frontiers Center for Brain Science, Fudan University
[3] Key Laboratory of Computational Neuroscience
and Brain-Inspired Intelligence, Fudan University
[4] State Key Laboratory of Medical Neurobiology, Institutes of Brain Science, Fudan University
[5]Shanghai Artificial Intelligence Laboratory

`{zhangjd20,qxzhu16,wlin}@fudan.edu.cn`

## Abstract

Control problems are always challenging since they arise from the real-world systems where stochasticity and randomness are of ubiquitous presence. This naturally and urgently calls for developing efficient neural control policies for stabilizing not only the deterministic equations but the stochastic systems as well. Here, in order to meet this paramount call, we propose two types of controllers, viz., the exponential stabilizer (ES) based on the stochastic Lyapunov theory and the asymptotic stabilizer (AS) based on the stochastic asymptotic stability theory. The ES can render the controlled systems exponentially convergent but it requires a long computational time; conversely, the AS makes the training much faster but it can only assure the asymptotic (not the exponential) attractiveness of the control targets. These two stochastic controllers thus are complementary in applications. We also investigate rigorously the linear controller and the proposed neural stochastic controllers in both convergence time and energy cost and numerically compare them in these two indexes. More significantly, we use several representative physical systems to illustrate the usefulness of the proposed controllers in stabilization of dynamical systems.

## 1 Introduction

In the field of controlling dynamical systems, one of the major missions is to find efficient control policies for stabilizing ordinary differential equations (ODEs) to targeted equilibriums. The policies for stabilizing linear or polynomial dynamical systems have been fully developed using the standard Lyapunov stability theory, e.g., the linear quadratic regulator (LQR) Khalil (2002) and the sum-of-squares (SOS) polynomials through the semi-definite planning (SDP) Parrilo (2000). As for stabilizing more general and nonlinear dynamical systems, linearization technique around the targeted states is often utilized and thus the existing control policies are effective in the vicinity of the targeted states (Sastry & Isidori, 1989) but likely lose efficacy in the region far away from those states. Moreover, in real applications, the explicit forms of the controlled nonlinear systems are often partially or completely unknown, so it is very difficult to directly design controllers only using the Lyapunov stability theory. To overcome these difficulties, designing the controllers via training neural networks (NNs) become one of the mainstream approaches in the community of cybernetics (Polycarpou, 1996). Recent outstanding developments using NNs include enlarging the safe region

---

*To whom correspondence should be addressed: Q.Z. and W.L,
https://faculty.fudan.edu.cn/wlin/zh_CN/index.htm.

(Richards et al., 2018), learning the stable dynamics (Takeishi & Kawahara, 2021), and constructing the Lyapunov function and the control function simultaneously (Chang et al., 2019). In Kolter & Manek (2019) a projected NN has been constructed to directely learn a stable dynamical system and fits the observed time series data well, but it did not focus on learning a control policy to stabilize the original dynamics. All these existing developments are formulated only for deterministic systems but inapplicable directly to the dynamical systems described by stochastic differential equations (SDEs), requiring us to include the stochasticity appropriately into the use of neural controls to different types of dynamical systems.

The stability theory for stochastic systems has been systematically developed in the past several decades. Representative contributions in the literature include the Lyapunov-like stability theory for SDEs Mao (2007), stabilization of unstable states in ODEs only using noise perturbations Mao (1994b), and the stability induced by randomly switching structures Guo et al. (2018). Generally, for any SDEs governed by $d\boldsymbol{x} = f(\boldsymbol{x})dt + g(\boldsymbol{x})dB_t$, control policies as $\boldsymbol{u} = (\boldsymbol{u}_f, \boldsymbol{u}_g)$ are introduced, which transforms the original equations into the controlled system $d\boldsymbol{x} = [f(\boldsymbol{x}) + \boldsymbol{u}_f(\boldsymbol{x})]dt + [g(\boldsymbol{x}) + \boldsymbol{u}_g(\boldsymbol{x})]dB_t$. Appropriate forms of control policies are able to steer the controlled system to the equilibriums that are unstable in the original SDEs. Traditional control methods focus on designing deterministic control $\boldsymbol{u}_f$ and regard noise as negative part. Innovatively, we treat noise as a beneficial part and design stochastic control $\boldsymbol{u}_g$ to achieve the stabilization.

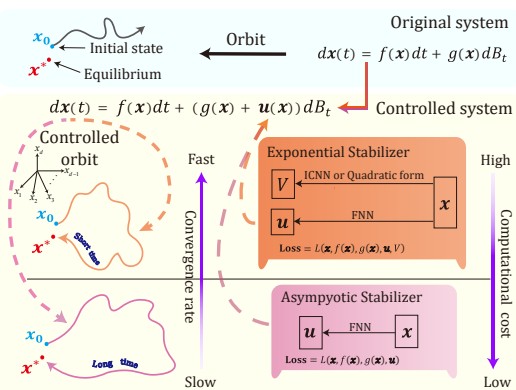

Figure 1: Sketches of the two frameworks of neural stochastic controller. Both the ES and AS find control function $\boldsymbol{u}$ with fully connected feedforward NN (FNN).

In this article, we articulate two frameworks of neural stochastic control which can complement each other in terms of convergence rate and the computational time of training NNs. Additionally, we analytically investigate the convergence time and the energy cost for the classic linear control and the proposed neural stochastic control and numerically compare them. We further extend our frameworks to model-free case with existing data reconstruction methods. The major contributions of this article are multi-folded, including:

- designing two frameworks of neural stochastic control, viz., the ES and the AS, and presenting their advantages in the stochastic control,
- providing theoretical estimation for ES/AS and classic linear control in terms of convergence time and the energy cost,
- computing the convergence time and the energy cost of particular stochastic neural control,
- demonstrating the efficacy of the proposed stochastic neural control in important control problems arising from representative physical systems, and we make our code available at `https://github.com/jingddong-zhang/Neural-Stochastic-Control`.

## 1.1 Related Works

● **Lyapunov Method in Machine Learning**    The recent work Chang et al. (2019) proposed an NN framework of learning the Lyapunov function and the linear control function simultaneously for stabilizing ODEs. In comparison, we select several specific types of NNs which have typical properties of the Lyapunov function. For instance, we use the input convex neural network (ICNN) Amos et al. (2017), constructing a positively definite convex function as a neural Lyapunov function (Kolter & Manek, 2019; Takeishi & Kawahara, 2021), and we construct the NN in a quadratic form (Richards et al., 2018; Gallieri et al., 2019) for linear or sublinear systems where the SDP method is often used to find the SOS-type Lyapunov function (Henrion & Garulli, 2005; Jarvis-Wloszek et al., 2003; Parrilo, 2000).

● **Stochastic Stability Theory of SDEs**    Stochastic stability theory for SDEs have been systematically and fruitfully achieved in the past several decades (Kushner, 1967; Arnold, 2007; Mao, 1991, 1994a). The positive effects of stochasticity have also been cultivated in control fields (Mao

et al., 2002; Deng et al., 2008; Caraballo et al., 2003; Appleby et al., 2008; Mao et al., 2007). These, therefore, motivate us to develop *only* neural stochastic control to stabilize different sorts of dynamical systems in this article. More stochastic stability theory for different kinds of systems are included in Appleby et al. (2006); Appleby (2003); Caraballo & Robinson (2004); Wang & Zhu (2017).

## 2 Preliminaries

To begin with, we consider the SDE which is written in a general form as:

$$\mathrm{d}\boldsymbol{x}(t) = F(\boldsymbol{x}(t))\mathrm{d}t + G(\boldsymbol{x}(t))\mathrm{d}B_t, \ \ t \geq 0, \ \boldsymbol{x}(0) = \boldsymbol{x}_0 \in \mathbb{R}^d, \tag{1}$$

where $F : \mathbb{R}^d \to \mathbb{R}^d$ is the drift function, and $G : \mathbb{R}^d \to \mathbb{R}^{d \times r}$ is the diffusion function with $\mathbb{R}^{d \times r}$, a space of $d \times r$ matrices with real entries, and $B_t \in \mathbb{R}^r$, a $r$-dimensional ($r$-D) Brownian motion. Without loss of generality, we set $F(\boldsymbol{0}) = \boldsymbol{0}$ and $G(\boldsymbol{0}) = \boldsymbol{0}$ so that $\boldsymbol{x}_0 = \boldsymbol{0}$ is a zero solution Eq. (1).

**Notations.** Denote by $\|\cdot\|$ the $L^2$-norm for any given vector in $\mathbb{R}^d$. Denote by $|\cdot|$ the absolute value of a scalar number or the modulus length of a complex number number. For $A = (a_{ij})$, a matrix of dimension $d \times r$, denote by $\|A\|_{\mathrm{F}}^2 = \sum_{i=1}^d \sum_{j=1}^r a_{ij}^2$ the Frobenius norm.

**Assumption 2.1** *(Locally Lipschitzian Continuity) For every integer $n \geq 1$, there is a number $K_n > 0$ such that*

$$\|F(\boldsymbol{x}) - F(\boldsymbol{y})\| \leq K_n \|\boldsymbol{x} - \boldsymbol{y}\|, \ \|G(\boldsymbol{x}) - G(\boldsymbol{y})\|_{\mathrm{F}} \leq K_n \|\boldsymbol{x} - \boldsymbol{y}\|,$$

*for any $\boldsymbol{x}, \boldsymbol{y} \in \mathbb{R}^d$ with $\|\boldsymbol{x}\| \vee \|\boldsymbol{y}\| \leq n$.*

**Definition 2.1** *(Derivative Operator) Define the differential operator $\mathcal{L}$ associated with Eq. (1) by*

$$\mathcal{L} \triangleq \sum_{i=1}^d F_i(\boldsymbol{x}) \frac{\partial}{\partial x_i} + \frac{1}{2} \sum_{i,j=1}^d [G(\boldsymbol{x})G^\top(\boldsymbol{x})]_{ij} \frac{\partial^2}{\partial x_i \partial x_j}.$$

**Definition 2.2** *(Exponential Stability) The zero solution of Eq. (1) is said to be almost surely exponentially stable, if $\limsup_{t \to \infty} \frac{1}{t} \log \|\boldsymbol{x}(t; \boldsymbol{x}_0)\| < 0$ a.s. for all $\boldsymbol{x}_0 \in \mathbb{R}^d$. Here and throughout, a.s. stands for the abbreviation of almost surely.*

Then, the following Lyapunov stability theorem will be used in the establishment of our main results.

**Theorem 2.2** *Mao (2007) Suppose that Assumptions 2.1 holds. Suppose further that there exist a function $V \in C^2(\mathbb{R}^d; \mathbb{R}_+)$ with $V(\boldsymbol{0}) = 0$, constants $p > 0$, $c_1 > 0$, $c_2 \in \mathbb{R}$ and $c_3 \geq 0$, such that (i) $c_1 \|\boldsymbol{x}\|^p \leq V(\boldsymbol{x})$, (ii) $\mathcal{L}V(\boldsymbol{x}) \leq c_2 V(\boldsymbol{x})$, and (iii) $|\nabla V^\top(\boldsymbol{x})G(\boldsymbol{x})|^2 \geq c_3 V^2(\boldsymbol{x})$ for all $\boldsymbol{x} \neq 0$ and $t \geq 0$. Then,*

$$\limsup_{t \to \infty} \frac{1}{t} \log \|\boldsymbol{x}(t; t_0, \boldsymbol{x}_0)\| \leq -\frac{c_3 - 2c_2}{2p} \ \ a.s.. \tag{2}$$

*In particular, if $c_3 - 2c_2 > 0$, the zero solution of Eq. (1) is exponentially stable almost surely.*

The following asymptotic theorem also will be used in the establishment of our main results.

**Theorem 2.3** *Appleby et al. (2008) Suppose that Assumptions 2.1 holds. Suppose further $\min_{\|\boldsymbol{x}\|=M} \|\boldsymbol{x}^\top G(\boldsymbol{x})\| > 0$ for any $M > 0$ and there exists a number $\alpha \in (0,1)$ such that*

$$\|\boldsymbol{x}\|^2 (2\langle \boldsymbol{x}, F(\boldsymbol{x}) \rangle + \|G(\boldsymbol{x})\|_{\mathrm{F}}^2) - (2-\alpha)\|\boldsymbol{x}^\top G(\boldsymbol{x})\|^2 \leq 0, \ \ \forall \boldsymbol{x} \in \mathbb{R}^d. \tag{3}$$

*Then, the unique and global solution of Eq. (1) satisfies $\lim_{t \to \infty} \boldsymbol{x}(t, \boldsymbol{x}_0) = \boldsymbol{0}$ a.s., and we call this property as asymptotic attractiveness.*

## 3 Designing Stable Stochastic Controller

Here, we assume that the zero solution of the following SDE:

$$\mathrm{d}\boldsymbol{x} = f(\boldsymbol{x})\mathrm{d}t + g(\boldsymbol{x})\mathrm{d}B_t \tag{4}$$

is unstable. Note that, for any nontrivial targeted equilibrium $\boldsymbol{x}^*$, a direct transformation $\boldsymbol{y} = \boldsymbol{x} - \boldsymbol{x}^*$ can make the zero solution as the equilibrium of the transformed system. Thus, our mission is to

stabilize the zero solution only. As such, we are to use the NNs to design the control $\boldsymbol{u} : \mathbb{R}^d \to \mathbb{R}^{d \times r}$ with $\boldsymbol{u}(\mathbf{0}) = \mathbf{0}$ and apply it to Eq. (4) as

$$\mathrm{d}\boldsymbol{x} = f(\boldsymbol{x})\mathrm{d}t + [g(\boldsymbol{x}) + \boldsymbol{u}(\boldsymbol{x})]\mathrm{d}B_t. \tag{5}$$

Since $\boldsymbol{u}$ is integrated with $\mathrm{d}B_t$ in the controlled system (5), we regard it as a stochastic controller. In what follows, two frameworks of neural stochastic control, the exponential stabilizer (ES) and the asymptotic stabilizer (AS), are articulated, respectively, in Sections 3.1 and 3.2. All these control policies are intuitively depicted in Figure 1.

## 3.1 Exponential Stabilizer

Once we find the Lyapunov function $V$ and the neural controller $\boldsymbol{u}$, making the controlled system (5) meet all the conditions assumed in Theorem 2.2, the equilibrium $\mathbf{0}$ can be exponentially stabilized. To this end, we first provide two different types of functions for constructing $V$, which actually could be complementary in applications. Then, we design the explicit forms of control function and loss function.

**ICNN $V$ Function.** We use the ICNN (Amos et al., 2017) to represent the candidate Lyapunov function $V$. This guarantees $V$ as a convex function with respect to the input $\boldsymbol{x}$. In order to further make $V$ as a true Lyapunov function, we use the following form:

$$\begin{aligned} &\boldsymbol{z}_1 = \sigma_0(W_0\boldsymbol{x} + b_0), \ \ \boldsymbol{z}_{i+1} = \sigma_i(U_i\boldsymbol{z}_i + W_i\boldsymbol{x} + b_i), \\ &g(\boldsymbol{x}) \equiv \boldsymbol{z}_k, \ \ i = 1, \cdots, k-1, \\ &V(\boldsymbol{x}) = \sigma_{k+1}(g(\mathcal{F}(\boldsymbol{x})) - g(\mathcal{F}(\mathbf{0}))) + \varepsilon\|\boldsymbol{x}\|^2, \end{aligned} \tag{6}$$

as introduced in Manek & Kolter (2020). Here, $W_i, b_i$ are real-valued weights, $U_i$ are positive weights, $\sigma_i$ are convex, monotonically non-decreasing activation functions in the $i$-th layer, $\varepsilon$ is a small positive constant, and $\mathcal{F}$ is a continuously differentiable and invertible function. In our framework, we require $V \in C^2(\mathbb{R}^d; \mathbb{R}_+)$ according to Definition 2.1; however, each activation function $\sigma_i \equiv \sigma$ in Manek & Kolter (2020) is $C^1$ only. Thus, we modify the original function as:

$$\sigma(x) = \begin{cases} 0, & \text{if } x \leq 0, \\ (2dx^3 - x^4)/2d^3, & \text{if } 0 < x \leq d, \\ x - d/2, & \text{otherwise} \end{cases} \tag{7}$$

Figure 2: The smoothed **ReLU** $\sigma(\cdot)$.

which not only approximates the typical **ReLU** activation but also becomes continuously differentiable to the second order (see Figure 2).

**Quadratic $V$ Function.** For any $\boldsymbol{x} \in \mathbb{R}^d$, let $V_{\boldsymbol{\theta}} \in \mathbb{R}^d$ be a multilayered feedforward NN of the input $\boldsymbol{x}$ and with $\tanh(\cdot)$ as the activation functions, where $\boldsymbol{\theta}$ is the parameter vector. To meet the condition used in Definition 2.1, we cannot use the **ReLU**, a non-smooth function, as the activation function. Hence, we use the candidate Lyapunov function as:

$$V(\boldsymbol{x}) = \boldsymbol{x}^\top \left[ \varepsilon I + V_{\boldsymbol{\theta}}(\boldsymbol{x})^\top V_{\boldsymbol{\theta}}(\boldsymbol{x}) \right] \boldsymbol{x}, \tag{8}$$

which was introduced in Gallieri et al. (2019). Here, $\varepsilon$ is a small positive constant.

**Control Function.** We introduce a multi-layer feedforward NN (FNN), denoted by $\mathbf{NN}(\boldsymbol{x}) \in \mathbb{R}^r$, to design the controller $\boldsymbol{u}$. Since we require $\boldsymbol{u}(\mathbf{0}) = \mathbf{0}$, we set $\boldsymbol{u}(\boldsymbol{x}) \triangleq \mathbf{NN}(\boldsymbol{x}) - \mathbf{NN}(\mathbf{0})$ or $\boldsymbol{u}(\boldsymbol{x}) \triangleq \mathrm{diag}(\boldsymbol{x})\mathbf{NN}(\boldsymbol{x})$ with $r = d$. Here, $\mathrm{diag}(\boldsymbol{x})$ is a diagonal matrix with its $i$-th diagonal element as $x_i$.

**Remark 3.1** *As reported in Chang et al. (2019), a single-layer NN without the bias constants in its arguments, which degenerates as linear control, could sufficiently take effect in the stabilization of many deterministic systems. However, this is NOT always the case for achieving the stabilization of highly nonlinear systems or even SDEs. The following proposition with Figure 3 provides an example, where neither the classic linear controller nor the stochastic linear controller can stabilize the unstable equilibrium in a particular SDE. The proof of this proposition is included in Appendix A.1.*

**Proposition 3.2** *Consider the following* 1*-D SDE:*

$$\mathrm{d}x(t) = x(t)\log|x(t)|\mathrm{d}t + u(x(t))\mathrm{d}B_t, \tag{9}$$

*with a zero solution $x^* = 0$. Then, for $u(x) = kx$ with any $k$ and $x_0 \neq 0$, $x^* = 0$ is neither exponentially stable nor of globally asymptotic attractiveness almost surely. For $u(x) = 2x^2$, $x^* = 0$ is of globally asymptotic attractiveness. For $u(x) \equiv 0$, the deterministic system cannot be stabilized by any classic linear controller.*

**Loss Function.** When the learning procedure updates the parameters in the NNs such that the constructed $V$ and $\boldsymbol{u}$ with the coefficient functions, $f$ and $g_{\boldsymbol{u}} \triangleq g + \boldsymbol{u}$, in the controlled system (5) meet all the conditions assumed in Theorem 2.2, the exponential stability of the controlled system is assured. Thus, we demand

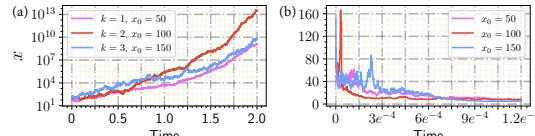

Figure 3: (a) $u(x) = kx$, (b) $u(x) = 2x^2$.

a suitable loss function to evaluate the likelihood that those conditions are satisfied. First, from Theorem 2.2, it follows that $V(\boldsymbol{x}) \geq \varepsilon\|\boldsymbol{x}\|^2$ for all $\boldsymbol{x} \in \mathbb{R}^d$. Thus, Conditions (ii)-(iii) together with $c_3 - 2c_2 > 0$ in Theorem 2.2 equivalently become

$$\inf_{\boldsymbol{x} \neq 0} \frac{(\nabla V(\boldsymbol{x})^\top g_{\boldsymbol{u}}(\boldsymbol{x}))^2}{V(\boldsymbol{x})^2} \geq b \cdot \sup_{\boldsymbol{x} \neq 0} \frac{\mathcal{L}V(\boldsymbol{x})}{V(\boldsymbol{x})}, \quad b > 2. \tag{10}$$

These conditions further imply that

$$\frac{(\nabla V(\boldsymbol{x})^\top g_{\boldsymbol{u}}(\boldsymbol{x}))^2}{V(\boldsymbol{x})^2} - b \cdot \frac{\mathcal{L}V(\boldsymbol{x})}{V(\boldsymbol{x})} \geq 0, \quad b > 2, \ \boldsymbol{x} \neq 0. \tag{11}$$

With these reduced conditions, we design the ES loss function for the controlled system (5) as follows.

**Definition 3.1** *(ES loss)* Consider a candidate Lyapunov function $V$ and a controller $\boldsymbol{u}$ for the controlled system (5). Then, the ES loss is defined as

$$L_{\mu,b,\varepsilon}(\boldsymbol{\theta}, \boldsymbol{u}) = \mathbb{E}_{\boldsymbol{x} \sim \mu}\Big[\max\Big(0, \frac{b \cdot \mathcal{L}V(\boldsymbol{x})}{V(\boldsymbol{x})} - \frac{(\nabla V(\boldsymbol{x})^\top g_{\boldsymbol{u}}(\boldsymbol{x}))^2}{V(\boldsymbol{x})^2}\Big)\Big],$$

where the state variable $\boldsymbol{x}$ obeys the distribution $\mu$. In practice, we consider the following empirical loss function:

$$L_{N,b,\varepsilon}(\boldsymbol{\theta}, \boldsymbol{u}) = \frac{1}{N}\sum_{i=1}^{N}\max\Big(0, \frac{b \cdot \mathcal{L}V(\boldsymbol{x}_i)}{V(\boldsymbol{x}_i)} - \frac{(\nabla V(\boldsymbol{x}_i)^\top g_{\boldsymbol{u}}(\boldsymbol{x}_i))^2}{V(\boldsymbol{x}_i)^2}\Big), \tag{12}$$

where $\{\boldsymbol{x}_i\}_{i=1}^{N}$ are sampled from the distribution $\mu = \mu(\Omega)$ and $\Omega$ is some closed domain in $\mathbb{R}^d$.

For convenience, we summarize the developed framework in Algorithm 1. Here, $b$ is a hyperparameter that can be adjusted as required by solving a specific problem.

**Remark 3.3** *In Section 5, we show numerically that the conditions reduced in* (11) *are sufficiently effective for designing the ES loss. Actually, it is not necessary to design the loss function using the conditions in* (10).

Now, for controlling any nonlinear ODEs or SDEs, we design the ES according to Algorithm 1. As such, using the ES framework can not only stabilize those unstable equilibriums (constant states) of the given systems, but also can stabilize those unstable oscillators, e.g., the limit cycle. This is because the solution corresponding to the oscillator can be regarded as a zero solution of the controlled system after appropriate transformations are implemented.

Another point needs attention. During the construction of $V$ in (6), $\varepsilon\|\boldsymbol{x}\|^2$, the $L^2$-regularization, is used to guarantee the positive definiteness of $V$. However, often in the application of the Lyapunov stability theory, the form of $\|\boldsymbol{x}\|^2$ is not always a suitable candidate for the Lyapunov function. It may restrict the generalizability of using our framework, so it needs necessary adjustments. The following example illustrates this point.

**Example 3.4** *Consider a 2-D SDE as follows:*

$$\begin{cases} \mathrm{d}x_1(t) = x_2(t)\mathrm{d}t, \\ \mathrm{d}x_2(t) = [-2x_1(t) - x_2(t)]\mathrm{d}t + x_1(t)\mathrm{d}B_t. \end{cases}$$

*In Appendix A.2, the zero solution of this system is validated to be exponentially stable almost surely; however, $k\|\boldsymbol{x}\|^2$ for any $k \in \mathbb{R}$ cannot be a useful auxiliary function to identify the exact stability of the zero solution.*

To be candid, using the current framework takes a longer time for training and constructing the neural Lyapunov function. In the next subsection, we thus establish an alternative control framework that can reduce the training time.

## 3.2 Asymptotic Stabilizer

Here, in light of Theorem 2.3, we are to establish the second framework, the AS, for stabilizing the unstable equilibrium of system (5). This framework only makes the equilibrium asymptotically attractive almost surely. Its control function is designed in the same way as the one used in the ES framework, whereas the loss function is differently designed.

**Definition 3.2** *(AS loss)* Utilization of the notations used in Definition 3.1, the loss function for the controlled system (5) with the controller $\boldsymbol{u}$ is defined as:

$$L_{\mu,\alpha}(\boldsymbol{u}) = \mathbb{E}_{\boldsymbol{x}\sim\mu}\big[\max\big(0, (\alpha-2)\|\boldsymbol{x}^\top g_{\boldsymbol{u}}(\boldsymbol{x})\|^2 + \|\boldsymbol{x}\|^2(\langle\boldsymbol{x}, f(\boldsymbol{x})\rangle + \|g_{\boldsymbol{u}}(\boldsymbol{x})\|_{\mathrm{F}}^2)\big)\big].$$

Akin to Definition 3.1, we set the empirical loss function as:

$$L_{N,\alpha}(\boldsymbol{u}) = \frac{1}{N}\sum_{i=1}^{N}\big[\max\big(0, (\alpha-2)\|\boldsymbol{x}_i^\top g_{\boldsymbol{u}}(\boldsymbol{x}_i)\|^2 + \|\boldsymbol{x}_i\|^2(\langle\boldsymbol{x}_i, f(\boldsymbol{x}_i)\rangle + \|g_{\boldsymbol{u}}(\boldsymbol{x}_i)\|_{\mathrm{F}}^2)\big)\big]. \quad (13)$$

Here, $\alpha$ is an adjustable parameter, which is related to the convergence time and the energy cost using the controller $\boldsymbol{u}$. We show in Appendix A.8 the influence of selecting different $\alpha$. For convenience, we summarize the AS framework in Algorithm 2.

## 4 Convergence Time and Energy Cost

The convergence time and the energy cost are the crucial factors to measure the quality of a controller (Yan et al., 2012; Li et al., 2017; Sun et al., 2017). In this section, we provide a comparative study between the traditional stochastic linear control and the ES/AS, the above-articulated neural stochastic control.

To this end, we first present a theorem on the estimations of the convergence time and the energy cost for the stochastic linear control on a general SDE.

**Theorem 4.1** *Consider the SDE with a stochastic linear controller as:*

$$\mathrm{d}\boldsymbol{x} = f(\boldsymbol{x})\mathrm{d}t + u(\boldsymbol{x})\mathrm{d}B_t, \quad \boldsymbol{x}(0) = \boldsymbol{x}_0 \in \mathbb{R}^d, \quad (14)$$

*where $\langle\boldsymbol{x}, f(\boldsymbol{x})\rangle \leq L\|\boldsymbol{x}\|^2$ and $u(\boldsymbol{x}) = k\boldsymbol{x}$ with $|k| > \sqrt{2L}$. Then, for $\epsilon < \|\boldsymbol{x}_0\|$, we have*

$$\begin{cases} \mathbb{E}[\tau_\epsilon] \leq T_\epsilon = \dfrac{2\log(\|\boldsymbol{x}_0\|/\epsilon)}{k^2 - 2L}, \\ \mathcal{E}(\tau_\epsilon, T_\epsilon) \leq \dfrac{k^2\|\boldsymbol{x}_0\|^2}{k^2 + 2L}\left[\exp\left(\dfrac{2(k^2 + 2L)\log(\|\boldsymbol{x}_0\|/\epsilon)}{k^2 - 2L}\right) - 1\right], \end{cases}$$

*where, for a sufficiently small $\epsilon > 0$, we denote the stopping time by $\tau_\epsilon \triangleq \inf\{t > 0 : \|\boldsymbol{x}(t)\| = \epsilon\}$ and denote the energy cost by*

$$\mathcal{E}(\tau_\epsilon, T_\epsilon) \triangleq \mathbb{E}\big(\int_0^{\tau_\epsilon \wedge T_\epsilon}\|\boldsymbol{u}\|^2\mathrm{d}t\big) = \mathbb{E}\big(\int_0^{T_\epsilon}\|\boldsymbol{u}\|^2\mathbb{1}_{\{t<\tau_\epsilon\}}\mathrm{d}t\big).$$

The proof of this theorem is provided in Appendix A.3.1.

We further consider the case for NN controller $u(x)$. In general, the $u(x)$ is Lipschitz continuous with Lipschitz constant $k_u$ under a suitable activation function such as **ReLU** in NN (Fazlyab et al., 2019; Pauli et al., 2021). Then we have the following upper bound estimation of the convergence time and the energy cost for ES and AS, whose proofs are provided in Appendix A.3.2, A.3.3.

**Theorem 4.2** *(Estimation for ES)* For ES stabilizer $u(x)$ in (14) with $\langle x, f(x) \rangle \leq L\|x\|^2$, $\varepsilon < \|x_0\|$, under the same notations and conditions in Theorem 2.2 with $c_3 - 2c_2 > 0$, we have

$$
\begin{cases}
\mathbb{E}[\tau_\epsilon] \leq T_\epsilon = \dfrac{2\log\left(V(x_0)/c_1\varepsilon^p\right)}{c_3 - 2c_2}, \\
\mathcal{E}(\tau_\epsilon, T_\epsilon) \leq \dfrac{k_u^2\|x_0\|^2}{k_u^2 + 2L}\left[\exp\left(\dfrac{2(k_u^2 + 2L)\log\left(V(x_0)/c_1\varepsilon^p\right)}{c_3 - 2c_2}\right) - 1\right].
\end{cases}
$$

**Theorem 4.3** *(Estimation for AS)* For (14) with $\langle x, f(x) \rangle \leq L\|x\|^2$, $\varepsilon < \|x_0\|$, under the same notations and conditions in Theorem 2.3, if the left term in (3) further satisfies $\max_{\|x\| \geq \varepsilon} \|x\|^{\alpha-4}(\|x\|^2(2\langle x, f(x)\rangle + \|u(x)\|_F^2) - (2-\alpha)\|x^\top u(x)\|^2) = -\delta_\varepsilon < 0$, then for NN controller $u(x)$ with Lipschizt constant $k_u$, we have

$$
\begin{cases}
\mathbb{E}[\tau_\epsilon] \leq T_\epsilon = \dfrac{2\left(\|x_0\|^\alpha - \varepsilon^\alpha\right)}{\delta_\varepsilon \cdot \alpha}, \\
\mathcal{E}(\tau_\epsilon, T_\epsilon) \leq \dfrac{k_u^2\|x_0\|^2}{k_u^2 + 2L}\left[\exp\left(\dfrac{2(k_u^2 + 2L)\left(\|x_0\|^\alpha - \varepsilon^\alpha\right)}{\delta_\varepsilon \cdot \alpha}\right) - 1\right].
\end{cases}
$$

Based on the theoretical results for ES and AS, we can further analyze the effects of hyperparameters $b, \alpha$ and neural network structures on the convergence time and energy cost in the control process. There are some interesting phenomena such as the monotonicity of $T_\varepsilon$ along $\alpha$ for AS change with the relative relationships of $\|x_0\|$ and $\varepsilon$, this inspires us to select suitable $\alpha$ according to the specific problems. We leave more discussions in Appendix A.3.4.

Now, we numerically compare the performances of the linear controller $u(x) = kx$ and the AS on the convergence time and the energy cost of the controls applied to system (14) with specific configurations (see Figure 4). We numerically find that $u(x) = kx$ can efficiently stabilize the equilibrium for $k > k_c = 5.6$. Without loss of generality, we fix $k = 6.0$, and compare the corresponding performances. As clearly shown in Figure 4, the AS outperforms $u(x) = kx$ from both perspectives, the convergence speed and the energy cost. In the simulations, the energy cost $\mathcal{E}(\tau_\epsilon, T_\epsilon)$ defined above is computed in a finite-time duration as $\mathcal{E}(\tau_\epsilon, T \wedge T_\epsilon)$, where $T < \infty$ is selected to be appropriately large. We leave more results of the comparison study in Appendix A.5.5.

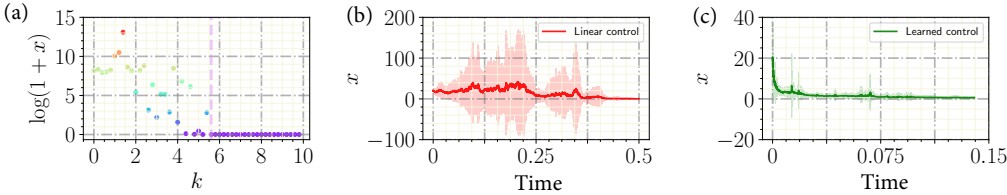

Figure 4: The performances of system (14) with specific configurations: $f(x) = x\log(1+x)$. (a) $u(x) = kx, k = 0.2 \cdot j, j = 1, \cdots, 50$, plot $\log(1 + x(1))$ against $k$. (b) Linear controller with $k = 6.0$, $\mathcal{E}(\tau_{0.1}, 1) = 38,388$. (c) AS control with $\mathcal{E}(\tau_{0.1}, 1) = 1438$

## 5 Experiments

In this section, we demonstrate the efficacy of the above-articulated frameworks of stochastic neural control, the ES and the AS, on several representative physical systems. We also compare these two frameworks, highlighting their advantages and weaknesses. The detailed configurations for these experiments are included in Appendix A.5. Additional illustrative experiments are included in Appendix A.6.

### 5.1 Harmonic Linear Oscillator

First, consider the harmonic linear oscillator $\ddot{y} + 2\beta\dot{y} + w^2 y = 0$, where $w$ is the natural frequency and $\beta > 0$ is the damping coefficient representing the strength of the external force on the vibrator (Dekker, 1981). Although this system is exponentially stable, the system with stochastic perturbations $\ddot{y} + (2\beta + \xi_2)\dot{y} + (w^2 + \xi_1)y = 0$ becomes unstable even if $\mathbb{E}[\xi_1(t)] = \mathbb{E}[\xi_2(t)] = 0$ (Arnold et al., 1983). Now, we apply the nonlinear ES(+ICNN), the linear ES(+Quadratic), and the nonlinear AS, respectively, to stabilizing this unstable dynamics (A.5.1) with $w^2 = 1$, $\beta = 0.5$, $\zeta_1 = -3$, and $\zeta_2 = 2.15$, the results are shown in Figure 5. Indeed, we find that

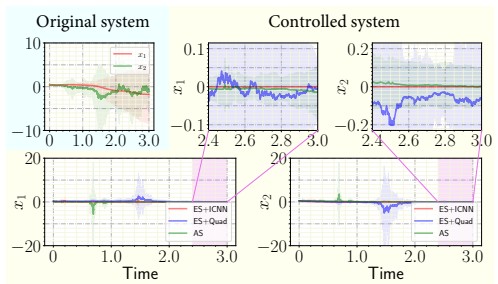

Figure 5: The solid lines are obtained through averaging the 20 sampled trajectories, while the shaded areas stand for the variance regions.

the two nonlinear stochastic neural controls are more robust than the linear control, and that the ES(+ICNN), rather than the AS, makes the controlled system more stable.

In Table 1 we compute the training time (Tt) used for the loss function converging to $0$, the number of iterations (Ni), the distance (Di) between the trajectory and the targeted equilibrium at time $T = 4$, and the convergence time (Ct) when the distance between the trajectory and the targeted equilibrium is less than $0.05$. The results are obtained through averaging the corresponding quantities produced by 20 randomly-sampled trajectories and the detailed training configurations are shown in Appendix A.5.1.

Table 1: Performance on Harmonic Linear Oscillator

|  | Tt | Ni | Di | Ct |
|---|---|---|---|---|
| ES(+ICNN) | 276.385s | 121 | **1e-9** | **0.459** |
| ES(+Quadratic) | 78.071s | **107** | 0.049 | 3.683 |
| AS | **4.839**s | 184 | 0.027 | 2.027 |

We further provide a comparison study of our newly proposed ES(+ICNN) with HDSCLF in (Sarkar et al., 2020), BALSA in (Fan et al., 2020) and classic LQR controller in controlling the harmonic linear oscillator. Both HDSCLF and BALSA are based on the Quadratic Program(QP), and they seek the control policy dynamically for each state in the control process. By contrast, our proposed learning control policy is directly

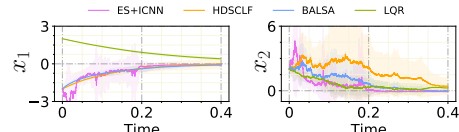

Figure 6: Comparison with existing methods.

used in the control process. Hence, our method is more efficient in the practical control problems. The results are shown in Figure. 6 (Please see more details in Appendix A.5.1). As can be seen that our learning control methods outperforms all others.

## 5.2 Stuart-Landau Equations

In this subsection, we show that our frameworks are beneficial to realizing the control and the synchronization of complex networks. To this end, we consider the single Stuart–Landau oscillator which is governed by the following complex-valued ODE:

$$\dot{Z} = (\beta + \mathrm{i}\gamma + \mu|Z|^2)Z, \ \ Z \in \mathbb{C}. \tag{15}$$

This equation is a paradigmatic model undergoing the so-called Andronov bifurcation (Kuznetsov, 2013): Stability of the equilibrium changes and the limit cycle emerges as the parameter passes some critical value. In what follows, we consider two cases based on system (15).

● **Case 1** We set $\beta = -25$, $\gamma = 1$, and $\mu = 1$, so that system (15) has a stable equilibrium $\rho = 0$ and an unstable limit cycle $\rho = 5$. Here, $Z = x + \mathrm{i}y = \rho e^{\mathrm{i}\theta}$. Now, the AS steers the dynamics to the unstable limit cycle, as successfully shown in Figure 7. The trajectories (the left column) and the phase orbits (the right column) of system (15), initiated from 30 randomly-selected initial states, without control (the upper panels) and with control of the AS (the lower panels). The initial values inside (resp., outside) the limit cycle $\rho = 5$ are indicated by the blue (resp., purple) pentagrams.

● **Case 2** Next, we consider the synchronization problem of the coupled Stuart-Landau equations. Successful deterministic methods have been systematically developed for realizing synchronization, including the adaptive control with time delay Selivanov et al. (2012) and the open-loop temporal network controller Zhang & Strogatz (2021). These methods majorly depend on the technique

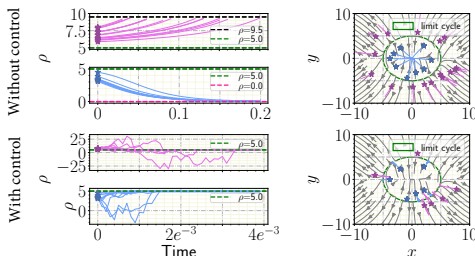

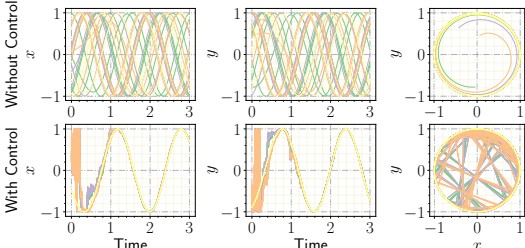

Figure 7: The trajectories (the left column) and the phase orbits (the right column) of system (15).

Figure 8: The dynamics of the first (resp., second) component of the coupled oscillators are shown in the panels in the left (resp., middle) column. The dynamics in the phase space (the right column).

of linearization in the vicinity of the synchronization manifold. Here, we show how to apply our framework to achieving the synchronization in the coupled system. We set the corresponding Laplace matrix $L = (L_{jk})_{n \times n}$ as $\sum_{k=1}^{n} L_{jk} = 0$, which guarantees the synchronization manifold is an invariant manifold of the coupled system (Pecora & Carroll, 1998). Specifically, we select as $n = 20$, $\sigma = 0.01$, $c_1 = -1.8$, $c_2 = 4$, and $L_{jk} = \delta_{jk} - \frac{1}{n}$, where $\delta_{jk}$ is the Kronecker function. Then, we apply the AS to this system and realize the stabilization of the synchronous manifold, as shown in Figure 8.

### 5.3 Data-Driven Pinning Control for Cell Fate Dynamics

Indeed, our frameworks can be extended to the model-free version via a combination with existing data reconstruction method. To be concrete, we show that our framework can combine with Neural ODEs (NODEs) (Chen et al., 2018) to learn the control policy from time series data for the Cell Fate system (Sun et al., 2017; Laslo et al., 2006), which describes the interaction be-

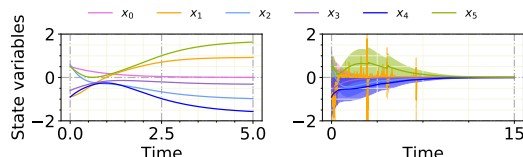

Figure 9: Pinning control for cell fate dynamics.

tween two suppressors during cellular differentiation for neutrophil and macrophage cell fate choices. The system $\dot{\boldsymbol{x}} = f(\boldsymbol{x})$, $\boldsymbol{x} = (x_1, ..., x_6)$ has three steady states: $P_{1,2,3}$, where $P_{2,3}$ correspond to different cell fates and are stable and $P_1$ represents a critical expression level connecting the two fates and is unstable. The network structure of this 6-D system is a treemap, where one root node $x_1$ can stabilize itself under the original dynamic. Hence, we choose root node $x_2$ with maximum our degree and add pinning control on it to stabilize the system to unstable state $P_1$. The original trajectory that converges to $P_2$ (left) and the controlled trajectory that converges to $P_1$ (right) are shown in Figure 17. The original trajectory is used to train the NODE to reconstruct the vector field $\hat{f}$, then we use the sample of $\hat{f}$ as training data to learn our stochastic pinning control. We provide experimental details in Appendix A.5.6.

In addition to the above controlled systems, we include the other illustrative examples, the controlled inverted pendulum, reservoir computing and the controlled Lorenz system, in Appendix A.5,A.6.

## 6 Conclusion and Future Works

In this article, we have proposed two frameworks of neural stochastic control for stabilizing different types of dynamical systems, including the SDEs. We have shown that the neural stochastic control outperforms the classic stochastic linear control in both the convergence time and the energy cost for typical systems. More importantly, using several representative physical systems, we have demonstrated the advantages of our frameworks and showed part of their weaknesses possibly emergent in real applications. Also, we present some limitations of the proposed frameworks in Appendix A.9. Moreover, we suggest several directions for further investigations: (i) acceleration of the training process of the ES, (ii) the basin stability of the neural stochastic control (Menck et al., 2013), (iii) the trade-off between the deterministic controller $\boldsymbol{u}_g$ using the NNs and the stochastic controller $\boldsymbol{u}_f$ using the NNs, (iv) the safe learning in Robotic control with small disturbances (Berkenkamp et al., 2017), and (v) the design of the purely data-driven stochastic neural control.

# 7 Acknowledgments

We thank the anonymous reviewers for their valuable and constructive comments that helped us to improve the work. Q.Z is supported by the Shanghai Postdoctoral Excellence Program (No. 2021091) and by the STCSM (Nos. 21511100200 and 22ZR1407300). W.L. is supported by the National Natural Science Foundation of China (No. 11925103) and by the STCSM (Nos. 19511101404, 22JC1402500, 22JC1401402, and 2021SHZDZX0103).

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
