# A  Appendix

## A.1  Proof of Proposition 3.2

First, we consider the solution of Eq. (9) for $u(x) = kx$. Applying Itô's formula to $\log |x|$ yields:

$$\begin{aligned}
\mathrm{d}\log |x| &= \frac{1}{|x|} \cdot \frac{x}{|x|} \mathrm{d}x - \frac{1}{2} \frac{1}{x^2} \mathrm{d}x \cdot \mathrm{d}x \\
&= \left( \log |x| - \frac{k^2}{2} \right) dt + k\mathrm{d}B_t.
\end{aligned}$$

By letting $y = \log |x| - \frac{1}{2}k^2$, we obtain $\mathrm{d}y = y\mathrm{d}t + k\mathrm{d}B_t$. Through applying Itô's formula of two dimensions to $e^{-t}y$, we get

$$\begin{aligned}
\mathrm{d}(e^{-t}y) &= \mathrm{d}(e^{-t})y + e^{-t}\mathrm{d}y + \mathrm{d}e^{-t}\mathrm{d}y \\
&= -e^{-t}y\mathrm{d}t + e^{-t}\mathrm{d}y - e^{-t}\mathrm{d}t\mathrm{d}y \\
&= ke^{-t}\mathrm{d}B_t,
\end{aligned}$$

which further implies

$$\begin{aligned}
y(t) &= e^t \left( y(0) + k \int_0^t e^{-s}\mathrm{d}B_s \right) \\
&\triangleq y(0)e^{-t} + ke^t\eta(t).
\end{aligned}$$

Here, the Gaussian process $\{\eta(t) = \int_0^t e^{-s}\mathrm{d}B_s,\ t \geq 0\}$ is a martingale. Thus, from the property of the martingale and Itô's isometry formula, it follows that

$$\begin{cases}
\mathbb{E}\eta(t) = \mathbb{E}\eta(0) = 0, \\
\mathbb{E}\eta(t)^2 = \mathbb{E}\int_0^t e^{-2s}\mathrm{d}s = \frac{1 - e^{-2t}}{2}.
\end{cases}$$

Hence, we have $y(t) \sim \mathcal{N}\left( y(0)e^t, \frac{k^2}{2}(e^{2t} - 1) \right)$, which further indicates that

$$\log |\boldsymbol{x}(t)| \sim \mathcal{N}\left( \left( \log |x(0)| - \frac{k^2}{2} \right) e^t + \frac{k^2}{2}, \frac{k^2}{2}\left(e^{2t} - 1\right) \right).$$

For any $k \in \mathbb{R}$, we choose $x(0)$ such that $\log |x(0)| - \frac{k^2}{2} > 0$. Thus, we have $\mathbb{P}(\log |x(t)| > 0) > 0$, which contradicts the asymptotic stability condition:

$$\lim_{t \to \infty} x(t) = 0 \ a.s. \iff \lim_{t \to \infty} \log |x(t)| = -\infty \ a.s..$$

Second, for $u(x) = 2x^2$, it is direct to validate that

$$x^2[2x^2 \log x + u(x)^2] - \frac{3}{2}x^2 u(x)^2 \leq 0.$$

This thus meets the conditions with $\alpha = \frac{1}{2}$ in Theorem 2.3. Then, in light of this theorem, the zero solution is of asymptotic attractiveness almost surely. Finally, for $u(x) \equiv 0$, direct calculations give a result that the classic linear controller cannot globally stabilize the deterministic system. Therefore, the proof is complete.

## A.2  Validation of Example 3.4

On one hand, we select as $V(\boldsymbol{x}) = k\|\boldsymbol{x}\|^2 \equiv k(x_1^2 + x_2^2)$ with $k > 0$, an undetermined coefficient. We thus get $\mathcal{L}V(\boldsymbol{x}) = k(x_1^2 - 2x_1x_2 - 2x_2^2)$ and $\nabla V^\top(\boldsymbol{x})g(\boldsymbol{x}) = 2kx_1x_2$. Notice that

$$\liminf_{(x_1^2+x_2^2)\neq 0} \frac{x_1^2 x_2^2}{(x_1^2 + x_2^2)^2} = 0.$$

So, to satisfy Condition (iii) in Theorem 2.2, we have to set $c_3 = 0$. As for meeting Condition (ii), letting $\boldsymbol{x} = (x, 0)$ gives $\mathcal{L}V = kx^2 > 0$, which indicates that, if there exists a number $c_2$ such that

$\mathcal{L}V(\boldsymbol{x}) \leq c_2 V(\boldsymbol{x})$, $c_2$ is positive. Hence, $c_3 - 2c_2 < 0$, so the above form of $V$ cannot guarantee the exponential stability of the zero solution.

On the other hand, we set as $\hat{V}(\boldsymbol{x}) \equiv \frac{5}{2}x_1^2 + x_1 x_2 + x_2^2$, and then we obtain

$$\mathcal{L}\hat{V}(\boldsymbol{x}) \leq -2\left[2x_1^2 + \frac{1}{2}x_2^2 + \frac{1}{2}(x_1 + x_2)^2\right] = -2\hat{V}(\boldsymbol{x}).$$

As we choose $c_2 = -2$ and $c_3 = 0$, all the conditions in Theorem 2.2 are satisfied. Therefore, the exponential stability of the zero solution is assured. This example particularly indicates that regularization terms need delicate design and fine-tune in applications.

### A.3  Proofs for Theorems

### A.3.1  Proof of Theorem 4.1

We first present an estimation for the upper bound of $\mathbb{E}[\tau_\epsilon]$. Applying Itô's formula to $\log \|\boldsymbol{x}\|$ yields:

$$\log \|\boldsymbol{x}\| - \log \|\boldsymbol{x}(0)\|$$
$$= \int_0^t \left(\frac{\langle \boldsymbol{x}(s), f(\boldsymbol{x}(s))\rangle}{\|\boldsymbol{x}\|^2} - \frac{k^2}{2}\right) \mathrm{d}s + \int_0^t k\mathrm{d}B_s$$
$$\leq \int_0^t \left(L - \frac{k^2}{2}\right) \mathrm{d}s + \int_0^t k\mathrm{d}B_s$$
$$= \left(L - \frac{k^2}{2}\right) t + kB_t.$$

Substitution of $t$ with a stopping time $\tau$ gives

$$\log \|\boldsymbol{x}(\tau)\| - \log \|\boldsymbol{x}_0\| \leq \left(L - \frac{k^2}{2}\right)\tau + kB_\tau,$$

which, after taking the expectation on both sides, yields:

$$\mathbb{E}\left[\log \frac{\|\boldsymbol{x}(\tau)\|}{\|\boldsymbol{x}_0\|}\right] \leq \mathbb{E}\left[\int_0^\tau \left(L - \frac{k^2}{2}\right)\mathrm{d}s\right].$$

Notice that $\boldsymbol{x}(\tau_\epsilon) = \epsilon$. Then, we have

$$\log \frac{\epsilon}{\|\boldsymbol{x}_0\|} = \mathbb{E}\left[\log \frac{\|\boldsymbol{x}(\tau_\epsilon)\|}{\|\boldsymbol{x}_0\|}\right] \leq \left(L - \frac{k^2}{2}\right)\mathbb{E}[\tau_\epsilon].$$

From $L - \frac{k^2}{2} < 0$ and $\log \frac{\epsilon}{\|\boldsymbol{x}_0\|} < 0$, it follows that

$$\mathbb{E}[\tau_\epsilon] \leq \frac{2\log\left(\frac{\|\boldsymbol{x}_0\|}{\epsilon}\right)}{k^2 - 2L} \triangleq T_\epsilon.$$

Next, by using the above results, we are in a position to provide an estimation of the energy $\mathcal{E}(\tau_\epsilon, T_\epsilon)$. To this end, an application of Itô's formula yields:

$$\|\boldsymbol{x}(t)\|^2 - \|\boldsymbol{x}(0)\|^2$$
$$= \int_0^t \left(2\langle \boldsymbol{x}(s), f(\boldsymbol{x}(s))\rangle + k^2\|\boldsymbol{x}(s)\|^2\right)\mathrm{d}s$$
$$+ \int_0^t 2k^2\|\boldsymbol{x}(s)\|^2\mathrm{d}B_s$$
$$\leq \int_0^t (2L + k^2)\|\boldsymbol{x}(s)\|^2\mathrm{d}s + \int_0^t 2k^2\|\boldsymbol{x}(s)\|^2\mathrm{d}B_s.$$

Thus, taking the expectation on both sides along the time interval $[0, t \wedge \tau_\epsilon]$ gives

$$\mathbb{E}[\|\boldsymbol{x}(t \wedge \tau_\epsilon)\|^2] \leq \|\boldsymbol{x}_0\|^2 + \mathbb{E}\int_0^{t\wedge\tau_\epsilon} (2L + k^2)\|\boldsymbol{x}(s)\|^2\mathrm{d}s$$
$$= \|\boldsymbol{x}_0\|^2 + (2L + k^2)\int_0^t \mathbb{E}[\|\boldsymbol{x}(s)\|^2 \mathbb{1}_{\{s<\tau_\epsilon\}}]\mathrm{d}s,$$

which further implies

$$\mathbb{E}[\|\boldsymbol{x}(t)\|^2 \mathbb{1}_{\{t<\tau_\epsilon\}}] \leq \mathbb{E}[\|\boldsymbol{x}(t \wedge \tau_\epsilon)\|^2]$$

$$\leq \|\boldsymbol{x}_0\|^2 + (2L + k^2) \int_0^t \mathbb{E}[\|\boldsymbol{x}(s)\|^2 \mathbb{1}_{\{s<\tau_\epsilon\}}]\mathrm{d}s.$$

Now, applying Gronwall's inequality, we get

$$\mathbb{E}[\|\boldsymbol{x}(t)\|^2 \mathbb{1}_{\{t<\tau_\epsilon\}}] \leq \|\boldsymbol{x}_0\|^2 e^{(2L+k^2)t}.$$

Finally, the energy is computed and estimated as follows:

$$\mathcal{E}(\tau_\epsilon, T_\epsilon) = \mathbb{E}\left( \int_0^{\tau_\epsilon \wedge T_\epsilon} k^2 \|\boldsymbol{x}(s)\|^2 \mathrm{d}s \right)$$

$$= k^2 \int_0^{T_\epsilon} \mathbb{E}[\|\boldsymbol{x}(t)\|^2 \mathbb{1}_{\{t<\tau_\epsilon\}}]\mathrm{d}s$$

$$\leq k^2 \|\boldsymbol{x}_0\|^2 \int_0^{T_\epsilon} e^{(2L+k^2)s} \mathrm{d}s$$

$$= \frac{k^2 \|\boldsymbol{x}_0\|^2}{2L + k^2} \left[ \exp(2L + k^2)T_\epsilon - 1 \right]$$

$$= \frac{k^2 \|\boldsymbol{x}_0\|^2}{k^2 + 2L} \left[ \exp\left( \frac{2(k^2 + 2L)\log\left( \frac{\|\boldsymbol{x}_0\|_2^2}{\epsilon} \right)}{k^2 - 2L} \right) - 1 \right].$$

This therefore completes the proof of the whole theorem.

### A.3.2 Proof of Theorem 4.2

First we prove the estimation for $\mathbb{E}[\tau_\varepsilon]$. Applying Itô's formula to $\log V(\boldsymbol{x})$ yields:

$$\log V(\boldsymbol{x}(t)) = \log V(\boldsymbol{x}_0) + \int_0^t \frac{\mathcal{L}V(\boldsymbol{x}(s))}{V(\boldsymbol{x}(s)} \mathrm{d}s + \int_0^t \frac{\nabla V(\boldsymbol{x}(s)) \cdot \boldsymbol{u}(\boldsymbol{x}(s))}{V(\boldsymbol{x}(s))} \mathrm{d}B_s$$

$$- \frac{1}{2} \int_0^t \frac{\|\nabla V \cdot \boldsymbol{u}\|^2}{V^2} \mathrm{d}s$$

$$\leq \log(V(\boldsymbol{x}_0)) + c_2 t + \int_0^t \frac{\nabla V(\boldsymbol{x}(s)) \cdot \boldsymbol{u}(\boldsymbol{x}(s))}{V(\boldsymbol{x}(s))} \mathrm{d}B_s - \frac{c_3 t}{2},$$

Substitution of t with a stopping time $\tau$ and taking expectation on both sides, we have

$$\mathbb{E}[\log V(\boldsymbol{x}(\tau))] \leq \mathbb{E}[\log V(\boldsymbol{x}_0)] + \frac{2c_2 - c_3}{2}\tau.$$

From $\|\boldsymbol{x}(\tau_\varepsilon)\| = \varepsilon < \|\boldsymbol{x}_0\|$ and $c_3 - 2c_2 > 0$, it follows that

$$\mathbb{E}[\tau_\varepsilon] \leq \frac{2\log(V(\boldsymbol{x}_0)/V(\boldsymbol{x}_{\tau_\varepsilon}))}{c_3 - 2c_2} \leq \min_{\|\boldsymbol{x}\|=\varepsilon} \frac{2\log(V(\boldsymbol{x}_0)/V(\boldsymbol{x}))}{c_3 - 2c_2} \leq \frac{2\log(V(\boldsymbol{x}_0)/c_1\varepsilon^p)}{c_3 - 2c_2} \triangleq T_\varepsilon.$$

Notice that NN control satisfies $\boldsymbol{u}(\boldsymbol{0}) = \boldsymbol{0}$, under the Lipschitz condition, we have $\|\boldsymbol{u}(\boldsymbol{x})\| \leq k_{\boldsymbol{u}}\|\boldsymbol{x}\|$. Then, similar to the procedure for the energy cost in A.3.1, we can get that

$$\mathbb{E}[\|\boldsymbol{x}(t)\|^2 \mathbb{1}_{\{t<\tau_\epsilon\}}] \leq \|\boldsymbol{x}_0\|^2 e^{(2L+k_{\boldsymbol{u}}^2)t},$$

$$\mathcal{E}(\tau_\varepsilon, T_\varepsilon) = \mathbb{E}\left( \int_0^{\tau_\varepsilon \wedge T_\varepsilon} \|\boldsymbol{u}(\boldsymbol{x}(s))\|^2 \mathrm{d}s \right)$$

$$\leq k_{\boldsymbol{u}}^2 \int_0^{T_\varepsilon} \mathbb{E}[\|\boldsymbol{x}(s)\|^2 \mathbb{1}_{\{s<\tau_\epsilon\}}]\mathrm{d}s$$

$$\leq \frac{k_{\boldsymbol{u}}^2 \|\boldsymbol{x}_0\|^2}{k_{\boldsymbol{u}}^2 + 2L} \left[ \exp\left( \frac{2(k_{\boldsymbol{u}}^2 + 2L)\log(V(\boldsymbol{x}_0)/c_1\varepsilon^p))}{c_3 - 2c_2} \right) - 1 \right],$$

which completes the proof.

### A.3.3 Proof of Theorem 4.3

First we prove the estimation for $\mathbb{E}[\tau_\varepsilon]$. Applying Itô's formula to $\|\boldsymbol{x}\|^2$ we have

$$\|\boldsymbol{x}(t)\|^2 = \|\boldsymbol{x}(0)\|^2 + \int_0^t (2\langle \boldsymbol{x}, f(\boldsymbol{x})\rangle + \|\boldsymbol{u}(\boldsymbol{x})\|^2)\mathrm{d}s + \int_0^t 2\langle \boldsymbol{x}, \boldsymbol{u}(\boldsymbol{x})\mathrm{d}B_s\rangle.$$

For $\alpha \in (0, 1)$, further using the fact that $\|\boldsymbol{x}\|^\alpha = (\|\boldsymbol{x}\|^2)^{\alpha/2}$ we have

$$\|\boldsymbol{x}(t)\|^\alpha = \|\boldsymbol{x}(0)\|^\alpha + \int_0^t \frac{\alpha}{2}\|\boldsymbol{x}\|^{\alpha-4}q(\boldsymbol{x})\mathrm{d}s + \int_0^t \alpha\|\boldsymbol{x}\|^{\alpha-2}\langle \boldsymbol{x}, \boldsymbol{u}(\boldsymbol{x})\mathrm{d}B_s\rangle,$$

$$q(\boldsymbol{x}) = \|\boldsymbol{x}\|^2(2\langle \boldsymbol{x}, F(\boldsymbol{x})\rangle + \|\boldsymbol{u}(\boldsymbol{x})\|_{\mathrm{F}}^2) - (2-\alpha)\|\boldsymbol{x}^\top\boldsymbol{u}(\boldsymbol{x})\|^2 \leq 0,$$

$$\max_{\|\boldsymbol{x}\|\geq\varepsilon} \frac{q(\boldsymbol{x})}{\|\boldsymbol{x}\|^{4-\alpha}} \leq -\delta_\varepsilon.$$

Notice that $\|\boldsymbol{x}(t)\| \geq \varepsilon$, $t \leq \tau_\varepsilon$, setting $t$ as $\tau_\varepsilon$ and taking expectation we have

$$\varepsilon^\alpha \leq \|\boldsymbol{x}_0\|^\alpha - \frac{\alpha}{2}\delta_\varepsilon\mathbb{E}[\tau_\varepsilon]$$

Then we have

$$\mathbb{E}[\tau_\varepsilon] \leq \frac{2(\|\boldsymbol{x}_0\|^\alpha - \varepsilon^\alpha)}{\alpha \cdot \delta_\varepsilon} \triangleq T_\varepsilon.$$

The estimation of the energy cost is just the same as that in A.3.2, and here we omit it.

### A.3.4 Discussions for Theorem 4.2,4.3

**Convergence Time for ES** From the conditions in Theorem 2.2 and the equivalent condition (10),(11), the hyperparameter $b$ for ES can be substantially regarded as $\frac{c_3}{c_2}$. Hence, when $b > 2$ we have $c_3 - 2c_2 > 0$, which corresponds to the exponential stability condition. This can be also confirmed by the upper bound estimation $\frac{2\log(V(\boldsymbol{x}_0)/c_1\varepsilon^p)}{c_3 - 2c_2} \triangleq T_\varepsilon$ in our Theorem 4.2, the convergence time $T_\varepsilon$ decreases as $c_3 - 2c_2$ get larger than zero, which means we should choose larger $b$ in the training. The $T_\varepsilon$ also decreases as the $c_1$ and $p$ increase, from the condition $V(\boldsymbol{x}) \geq c_1\|\boldsymbol{x}\|^p$ in Theorem 2.2 and the truth that $\nabla(c_1\|\boldsymbol{x}\|^p) = c_1p\|\boldsymbol{x}\|^{p-1}\nabla(\|\boldsymbol{x}\|)$, we can summarize that the slope of $V(\boldsymbol{x})$ can affect the convergence rate for ES. This inspire that we can design more steep neural networks structure for $V$, such as multiply $\|\boldsymbol{x}\|^\lambda$, $\lambda > 0$ to $V$ or increase $\varepsilon$ in $V$ in ES(+Quad).

**Convergence Time for AS** From the upper bound estimation $\frac{2(\|\boldsymbol{x}_0\|^\alpha - \varepsilon^\alpha)}{\alpha \cdot \delta_\varepsilon} \triangleq T_\varepsilon$ in Theorem 4.3, the convergence time is indeed related to $\alpha$. This parameter is not covered in the learning parameters, so we can adjust it according to the practical requirements. Here, we study the influence of $\alpha$ with fixed $\varepsilon$, that is, we consider the function $h(\alpha) = \frac{a^\alpha - b^\alpha}{\alpha}$, $a > b$, $\alpha \in (0, 1)$. The interesting thing is that monotonicity of $h(\alpha)$ changes with the parameters $a, b$, as shown in Figure 10. This phenomenon indicates that the minimal point for convergence time in AS should be selected according to the parameter region of $(\|\boldsymbol{x}_0\|, \varepsilon)$. The analytical or numerical forms of critical curve in this region need further research in future work.

### A.3.5 More Discussion of AS/ES

**Understanding AS Loss** We utilize the formula $\|\boldsymbol{x}\|^2(2\langle \boldsymbol{x}, F(\boldsymbol{x})\rangle + \|G(\boldsymbol{x})\|_{\mathrm{F}}^2) - (2 - \alpha)\|\boldsymbol{x}^\top G(\boldsymbol{x})\|^2 \triangleq q(\boldsymbol{x})$ in (3) to construct the AS loss. Here we explain this term in more detail. The $q(\boldsymbol{x})$ is derived from the chain derivative of $\|\boldsymbol{x}\|^\alpha = (\|\boldsymbol{x}\|^2)^{\alpha/2}$ according to Itô's formula, we then have $\mathrm{d}\|\boldsymbol{x}\|^\alpha = \alpha/2\|\boldsymbol{x}\|^{\alpha-4}q(\boldsymbol{x})\mathrm{d}t + \alpha\|\boldsymbol{x}\|^{\alpha-2}\|\boldsymbol{x}^\top G(\boldsymbol{x})\|\mathrm{d}B_t$. The drift term is less than zero when $q(\boldsymbol{x}) \leq 0$, which can guarantee the bounded existence and asymptotic stability of $\|\boldsymbol{x}\|^\alpha$ from the semi-martingale convergence theorem (Liptser & Shiryayev, 2012) .

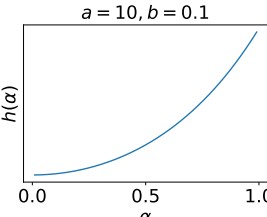 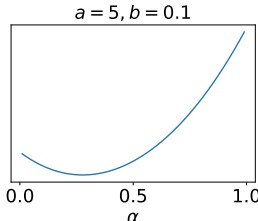 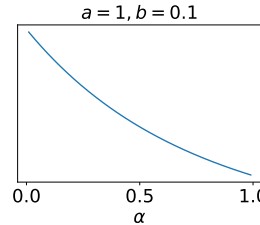

Figure 10: Plot for $h(\alpha)$ with different parameters combinations.

**Computational Complexity** The AS framework is computationally efficient because we only need the tensor operation in the training process. The computational complexity of this procedure is $\mathcal{O}(mn)$ for $m$ data on $n$-D dynamics. The training for ES framework is not as efficient as AS. The major reason is that we should compute the Hessian matrix of $V$ to get $\mathcal{L}V$ and the computational complexity of this operator is $\mathcal{O}(mn^2)$.

### A.4 Algorithms

We summarize the Algorithms of ES and AS as follows:

---
**Algorithm 1:** Exponential Stabilizer
---
**Input:** Data $\{x_i\}_{i=1}^n$ sampled from $\mu(\Omega)$, iteration step $m$, learning rate $\gamma$, training error $\delta$, coefficient samples $f(x_i)$ and $g(x_i)$, initial parameters $\theta_V(0), \theta_u(0)$, and parameters $\varepsilon$ used in (6) and $b$ used in (11).
**Output:** Controller $u_{\theta_u}(x_i)$ and Lyapunov function $V_{\theta_V}(x_i)$ in the form of (6) or (8).
**for** $r = 0$ **to** $m - 1$ **do**
    Compute $\nabla V(x_i), HV(x_i), \; i = 1, \cdots, n$
    Compute ES loss: $L(\theta_V(r), \theta_u(r))$ from (12)
    $\theta_u(r+1) = \theta_u(r) - \gamma \cdot \nabla_{\theta_u} L(\theta_V(r), \; \theta_u(r)), \; \theta_V(r+1) = \theta_V(r) - \gamma \cdot \nabla_{\theta_V} L(\theta_V(r), \; \theta_u(r))$
    ▷ Update parameters
    **if** $L(\theta_V(r+1), \theta_u(r+1)) \leq \delta$ **then**
        break
---

---
**Algorithm 2:** Asymptotic Stabilizer
---
**Input:** Data $\{x_i\}_{i=1}^n$ sampled from $\mu(\Omega)$, parameter $\alpha \in (0, 1)$ used in (13), and all other parameters, $m, \gamma, \delta, f(x_i)$, and $g(x_i)$, defined in the same manner as those in Algorithm 1.
**Output:** Controller $u_{\theta_u}(x_i)$.
**for** $r = 0$ **to** $m - 1$ **do**
    Compute loss function: $L(\theta_u(r))$ from (13)
    $\theta_u(r+1) = \theta_u(r) - \gamma \cdot \nabla_{\theta_u} L(\theta_u(r))$                 ▷ Update parameters
    **if** $L(\theta_u(r+1)) \leq \delta$ **then**
        break
---

### A.5 Experimental Configurations

In this section, we provide the detailed descriptions for the experimental configurations of the physical examples in the main text. The computing device that we use for calculating our examples includes a single i7-10870 CPU with 16GB memory, while the computational frameworks for our neural stochastic control contain three typical NNs.

1. The ICNN $V$ function is constructed as:

$$\boldsymbol{z}_1 = \sigma(W_0\boldsymbol{x} + b_0),$$
$$\boldsymbol{z}_{i+1} = \sigma(U_i\boldsymbol{z}_i + W_i\boldsymbol{x} + b_i),\ i = 1, \cdots, k-1,$$
$$p(\boldsymbol{x}) \equiv \boldsymbol{z}_k,$$
$$V(\boldsymbol{x}) = \sigma(p(\boldsymbol{x}) - p(\boldsymbol{0})) + \varepsilon\|\boldsymbol{x}\|^2,$$

where $\sigma$ is the smoothed **ReLU** function as defined in the main text, $W_i \in \mathbb{R}^{h_i \times d}$, $U_i \in (\mathbb{R}_+ \cup \{0\})^{h_i \times h_{i-1}}$, $\boldsymbol{x} \in \mathbb{R}^d$, and, for simplicity, this ICNN function is denoted by ICNN$(h_0, h_1, \cdots, h_{k-1})$;

2. The quadratic form $V$ function is constructed as:

$$\boldsymbol{z}_1 = \tanh(W_0\boldsymbol{x} + B_1),$$
$$\boldsymbol{z}_{i+1} = \tanh(W_i\boldsymbol{z}_i + b_i),\ i = 1, \cdots, k-1,$$
$$V_{\boldsymbol{\theta}}(\boldsymbol{x}) \equiv W_k\boldsymbol{z}_k,$$
$$V(\boldsymbol{x}) = \boldsymbol{x}^\top(\varepsilon I + V_{\boldsymbol{\theta}}(\boldsymbol{x})^\top V_{\boldsymbol{\theta}}(\boldsymbol{x}))\boldsymbol{x}.$$

where $W_i \in \mathbb{R}^{h_{i+1} \times h_i}$, and this quadratic function is denoted by Quadratic$(h_0, h_1, \cdots, h_{k+1})$;

3. The neural control function (nonlinear version) is constructed as:

$$\boldsymbol{z}_1 = \mathcal{F}(W_0\boldsymbol{x} + B_1),$$
$$\boldsymbol{z}_{i+1} = \mathcal{F}(W_i\boldsymbol{z}_i + b_i),\ i = 1, \cdots, k-1,$$
$$\mathbf{NN}(\boldsymbol{x}) \equiv W_k\boldsymbol{z}_k,$$
$$\boldsymbol{u}(\boldsymbol{x}) = \operatorname{diag}(\boldsymbol{x})\mathbf{NN}(\boldsymbol{x}),$$

where $\mathcal{F}(\cdot)$ is the activation function, $W_i \in \mathbb{R}^{h_{i+1} \times h_i}$, and this control function is denoted by Control$(h_0, h_1, \cdots, h_{k+1})$;
The neural control function (linear version) is set as $\boldsymbol{u}(\boldsymbol{x}) = W_1\boldsymbol{x}$.

### A.5.1 Controlling Harmonic Linear Oscillator

Mathematically, the perturbed harmonic oscillator is written as:

$$\begin{cases} \mathrm{d}x_1 = x_2\mathrm{d}t, \\ \mathrm{d}x_2 = \left(-w^2x_1 - 2\beta x_2\right)\mathrm{d}t + \left(\zeta_1 x_1 + \zeta_2 x_2\right)\mathrm{d}B_t. \end{cases}$$

The controlled harmonic linear oscillator with the random damping and the random restoring force is written as:

$$\mathrm{d}x_1 = x_2\mathrm{d}t + u_1(x_1, x_2)\mathrm{d}B(t),$$
$$\mathrm{d}x_2 = -\left(x_1 + x_2\right)\mathrm{d}t + \left[-3x_1 + 2.15x_2 + u_2(x_1, x_2)\right]\mathrm{d}B(t).$$

As introduced in the main text, we use three combinations of our neural control frameworks:

1. ES(+ICNN): $\varepsilon = 10^{-3}$, ICNN$(6, 6, 1)$, and Control$(2, 6, 6, 2)$ with $\mathcal{F}(\cdot) = $ **ReLU** and $b = 2.1$,

2. ES(+Quadratic): $\varepsilon = 10^{-2}$, Quadratic$(2, 6, 2)$, and the controller is set as $\boldsymbol{u}(\boldsymbol{x}) = W_1\boldsymbol{x}$ with $W_1 \in \mathbb{R}^{2 \times 2}$ and $b = 2.1$, and

3. AS: Control$(2, 6, 6, 2)$ with $\mathcal{F}(\cdot) = $ **ReLU** and $\alpha = 0.75$.

We train these NNs, respectively, for 500 iterations using the 500 data that are sampled from $\mathcal{U}([-6, 6]^2)$, and we depict their loss functions in Figure 11, respectively.

In the main text, we compare the performance of the three learning control functions using the same initial value $(0.3, 0.5)$ along the time interval $[0, 4]$. In Figure 12, we display the results along the interval $[0, 3]$, where we randomly select the initial values from $\mathcal{U}([-2, 2]^2)$ and sample the corresponding 20 trajectories using different neural controllers.
In comparison with existing methods, we use the above ES(+ICNN) control. For LQR controller, we select $Q = 20I_2$, $R = I_2$ to stabilize the state to zero, the corresponding feedback control matrix

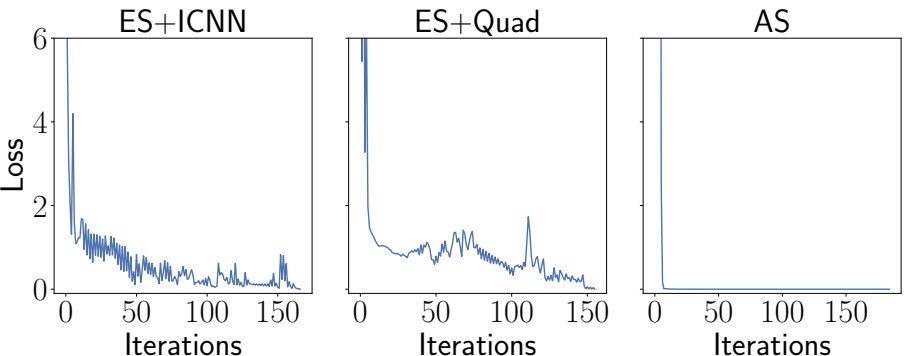

Figure 11: The Loss functions in the training processes, respectively, for using the ES(+ICNN) (left), the ES(+Quadratic) (middle), and the AS (right), respectively.

solved from the Riccati equation is $K = ((4.45, 0.09)^\top, (0.09, 3.6)^\top)$. For both the HDSCLF and BALSA method, there are three variables $u_1, u_2, d$ to be solved by Quadratic Program (QP), where $u_1, u_2$ are the control to be found and $d$ is the relaxation parameter. We pick the objective function as $u_1^2 + u_2^2 + 10d^2$, and select control Lyapunov function $V$ as the LQR solution K in HDSCLF, $V = \frac{1}{2}\boldsymbol{x}^\top P \boldsymbol{x}$ with $A^\top P + PA = -N$ for some $N > 0$ as required in BALSA (Fan et al., 2020), where $A$ is the drift matrix. The solution of this Lyapunov function with $N = ((2, 1)^\top, (1, 4)^\top)$ is $P = ((3, 1)^\top, (1, 3)^\top)$.

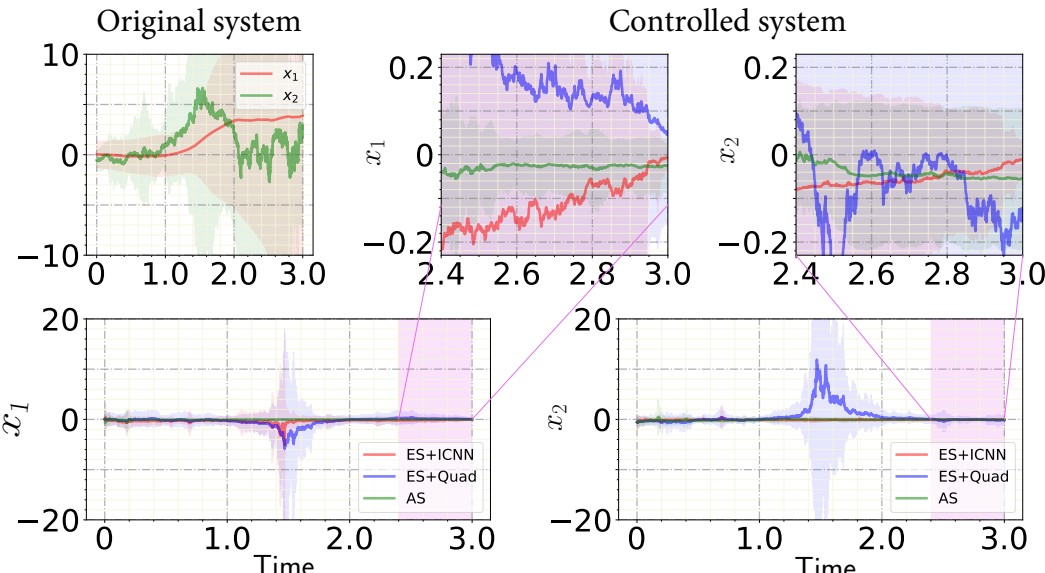

Figure 12: The 20 trajectories with the initial values sampled from $\mathcal{U}([-2, 2]^2)$. The trajectories of the controlled system are displayed along the two time intervals: $[0, 3.0]$ (bottom) and $[2.4, 3.0]$ (top middle & right). Here, the solid line and the shaded areas are the same as those defined in Figure 5.

### A.5.2 Controlling Inverted Pendulum

Controlling the inverted pendulum is one of the classic nonlinear control problems (Anderson, 1989; Huang & Huang, 2000), which is governed by $ml^2\ddot{\theta} = mgl\sin\theta - \beta\dot{\theta}$. Here, $m$ is the mass of the pendulum, $g$ is the gravitational acceleration, $l$ is the pole length, and $\beta$ is the friction coefficient. Mathematically, the pendulum can be written as a system with two state variables: $\theta$, the angle deviating from the vertical position $x = 0$, and $\dot{\theta}$, the angular velocity. Denote the 2-D state variable by $\boldsymbol{x} = (\theta, \dot{\theta})$. Then, we apply the AS articulated in Algorithm 2 to steer the system to the equilibrium

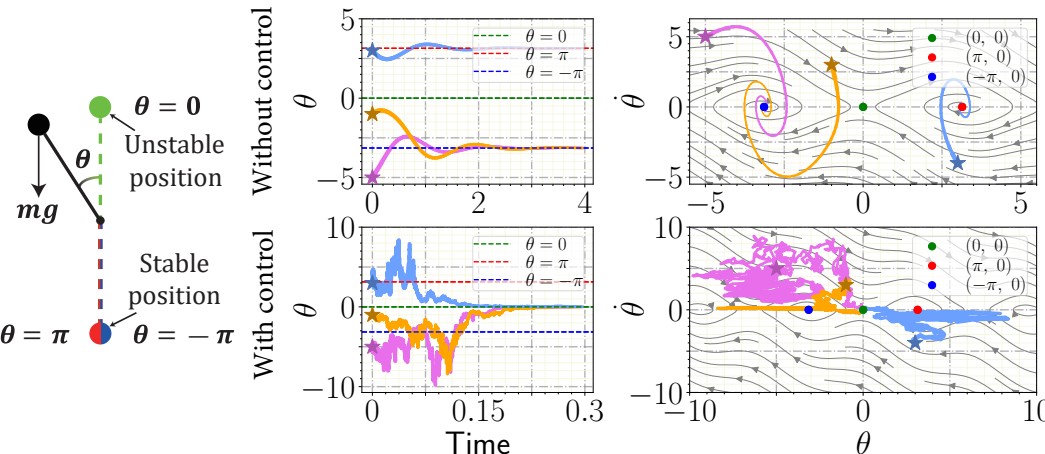

Figure 13: A schematic diagram of the inverted pendulum system (left), the sampled trajectories of the angle $\theta$ (middle), and the corresponding orbits in the phase space (right).

$x = 0$ (sustaining the pendulum inverted). The schematic diagram of the inverted pendulum system and successful control results are shown in Figure 13. The sampled trajectories are initiated from different angles and angular velocities [i.e., $(\theta(0), \dot{\theta}(0)) = (-5, 5), (-1, 3), (3, -4)$].

The controlled inverted pendulum equation is written as:

$$
\begin{aligned}
\mathrm{d}x &= y\mathrm{d}t + u_1(x, y)\mathrm{d}B_t, \\
\mathrm{d}y &= \frac{g}{l}\sin x - \frac{\beta}{ml^2}y + u_2(x, y)\mathrm{d}B_t.
\end{aligned}
\tag{16}
$$

Here, we use the AS to design the neural stochastic controllers $u_{1,2}$ to stabilize the system to the inverted position. As such, we select $\alpha = 0.5$ and Control$(2, 6, 2)$ with $\mathcal{F}(\cdot) = $ **ReLU**. We train the NNs using the $1,000$ data sampled from $\mathcal{U}([-10, 10]^2)$. A successful control has been shown in the main text (Figure 13). We perform further specific experiments on this system in Appendix A.8.

### A.5.3 Controlling Stuart-Landau Equation

The underlying system is written as $\dot{Z} = (-25 + \mathrm{i} + |Z|^2)Z$. Let $Z = \rho e^{\mathrm{i}\theta}$. Then, the original system is transformed into $\dot{\rho} = (\rho^2 - 25)\rho$ and $\dot{\theta} = 1$. Since $\rho = 5$ corresponds to the unstable limit cycle, we use the transformation $\tilde{\rho} = \rho - 5$ to shift it to the zero equilibrium, and then consider the following controlled system:

$$
\mathrm{d}\tilde{\rho} = \tilde{\rho}(\tilde{\rho} + 5)(\tilde{\rho} + 10)\mathrm{d}t + u(\tilde{\rho})\mathrm{d}B(t).
$$

Here, for constructing the AS, we use the nonlinear control function as Control$(1, 10, 1)$ with $\mathcal{F}(\cdot) = $ **Tanh** and $\alpha = 0.5$. Particularly, we train the NNs using the $3,000$ data sampled from $\mathcal{U}([-30, 30])$. We also sample 30 initial values from $(\rho, \theta) \sim \mathcal{U}([3, 10]) \times \mathcal{U}([0, 6.28])$ and numerically get their solutions without and with neural stochastic control along the time interval $[0, 0.2]$. Then, we use the constructed AS to stabilize the system successfully, as shown in the main text (Figure 7).

### A.5.4 Synchronization of Stuart-Landau Equations

Here, we consider the $n$ Stuart-Landau oscillators coupled in the form of

$$
\dot{Z}_j = Z_j - (1 + \mathrm{i}c_2)|Z_j|^2 Z_j - \sigma(1 + \mathrm{i}c_1)\sum_{k=1}^{n} L_{jk}Z_k,
$$

where $\sum_{k=1}^{n} L_{jk} = 0$ for all $j = 1, \cdots, n$. Using the coordinates $Z_j = \rho_j e^{i\theta_j}$, we have

$$
\begin{aligned}
\dot{\rho}_j =& \rho_j - \rho_j^3 - \sigma \sum_{k=1}^{n} L_{jk}\rho_k \left[\cos(\theta_k - \theta_j) - c_1 \sin(\theta_k - \theta_j)\right], \\
\dot{\theta}_j =& -c_2\rho_j^2 - \sigma \sum_{k=1}^{n} L_{jk}\frac{\rho_k}{\rho_j}\left[c_1 \cos(\theta_k - \theta_j) + \sin(\theta_k - \theta_j)\right].
\end{aligned}
\tag{17}
$$

Let $\tilde{\rho}_j = \rho_j - 1$ and notice that system (17) only depends on $\theta_k - \theta_j$ for $j, k = 1, \cdots, n$. Thus, letting $\tilde{\theta}_j = \theta_j - \theta_{j+1}$ yields:

$$
\begin{aligned}
\dot{\tilde{\rho}}_j =& (\tilde{\rho}_j + 1) - (\tilde{\rho}_j + 1)^3 - \sigma \sum_{k=1}^{n} L_{jk}(\tilde{\rho}_k + 1)p(\theta_k - \theta_j), \\
\dot{\tilde{\theta}}_j =& -c_2 \left[(\tilde{\rho}_j + 1)^2 - (\tilde{\rho}_{j+1} + 1)^2\right] \\
& - \sigma \sum_{k=1}^{n} \left[L_{jk}\frac{\tilde{\rho}_k + 1}{\tilde{\rho}_j + 1}q(\theta_k - \theta_j) - L_{j+1,k}\frac{\tilde{\rho}_k + 1}{\tilde{\rho}_{j+1} + 1}q(\theta_k - \theta_j)\right],
\end{aligned}
\tag{18}
$$

where

$$
p(\theta) = \cos\theta - c_1 \sin\theta, \quad q(\theta) = c_1 \cos\theta + \sin\theta,
\tag{19}
$$

$$
\theta_k - \theta_j = \begin{cases} -\tilde{\theta}_j - \cdots - \tilde{\theta}_{k-1}, \ k > j, \\ 0, \ k = j, \\ \tilde{\theta}_k + \cdots + \tilde{\theta}_{j-1}. \end{cases}
\tag{20}
$$

Hence, we successfully transform the original coupled system (17) with the states $\{\rho_j, \theta_j, \ j = 1, \cdots, n\}$ into a new coupled system (18) with (18)-(20) and the states $\{\tilde{\rho}_i, \ i = 1, \cdots, n; \ \tilde{\theta}_j, \ j = 1, \cdots, n-1\}$. In particular, the synchronous manifold $\{\rho_j = 1, \ \theta_1 = \cdots = \theta_n\}$ in the original system (17) becomes the equilibrium of the transformed system (18), i.e., $\{\tilde{\rho}_i = 0, \ \tilde{\theta}_j = 0\}$.

Now, we use the AS, the neural stochastic control, to stabilize this equilibrium of system (18) and equivalently realize the synchronization in the original coupled system. For implementing numerical experiment, we set $c_1 = -1.8$, $c_2 = 4$, $\sigma = 0.01$, and $n = 20$. We further set the Laplace matrix $L = (L_{jk})_{n \times n}$ as

$$
L_{jk} = \begin{cases} 1 - \dfrac{1}{n}, \ j = k, \\ -\dfrac{1}{n}, \ j \neq k. \end{cases}
\tag{21}
$$

We construct the AS as Control$(39, 80, 39)$ with $\mathcal{F}(\cdot) = \mathbf{ReLU}$ and $\alpha = 0.75$. Particularly, we sample $5,000$ points from

$$
(\tilde{\rho}_1, \cdots, \tilde{\rho}_{20}) \times (\tilde{\theta}_1, \cdots, \tilde{\theta}_{19}) \sim \mathcal{U}([0,5]^{20}) \times \mathcal{U}([-5,5]^{19}),
$$

and get the information of the dynamics based on system (18) with (19) and (20). The training process for the AS seems to be pretty efficient, as shown in Figure 14.

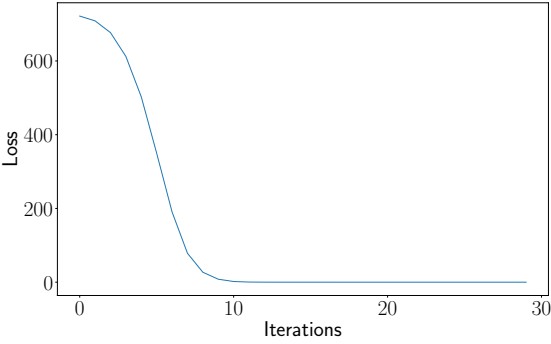

Figure 14: The Loss function for training the AS, dropping to 1.23e-5 within only 30 iterations.

As for generating the controlled trajectories for $\rho, \theta$, we make the following descriptions.

First, we randomly select the initial values for $\{\tilde{\rho}_i, \tilde{\theta}_j\}_{n=20}$ (we denote $\tilde{\theta}_{20}$ as $\theta_{20}$) from $(\mathcal{U}[0,5])^{20} \times (\mathcal{U}[-1,1])^{20}$, and thus obtain the initial values for $\{\rho_j, \theta_j\}_{n=20}$ based on (20). Secondly, without any control, we generate the trajectories using (17) and (18), respectively. Finally, after the neural stochastic control is implemented, we use $\rho = \tilde{\rho} + 1$ and $(\theta_1, \cdots, \theta_{20})^\top = \mathcal{A}(\tilde{\theta}_1, \cdots, \tilde{\theta}_{19}, \theta_{20})^\top$ to get the controlled trajectories for $\rho$ and $\theta$, where

$$\mathcal{A} = \begin{pmatrix} 1 & 1 & \cdots & 1 \\ 0 & 1 & \cdots & 1 \\ \vdots & \vdots & \vdots & \vdots \\ 0 & 0 & \cdots & 1 \end{pmatrix}_{20 \times 20} .$$

### A.5.5 Energy Cost of Control

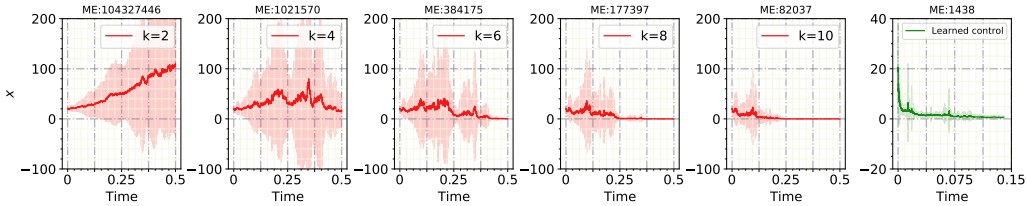

Figure 15: The convergence performance of different $k$ selected from $\{2, 4, 6, 8, 10\}$ and AS control. The mean energy(ME) $\mathcal{E}(\tau_{0.1}, 1.0)$ are computed from 20 random seeds $\{4r + 1, r = 0, \cdots, 19\}$

For the controlled nearly-linear SDE: $\mathrm{d}x = x \log(1 + x)\mathrm{d}t + u(x)\mathrm{d}B_t$, $x \in \mathbb{R}$. We construct the AS as Control$(1, 6, 1)$ with $\mathcal{F}(\cdot) = \mathbf{ReLU}$ and $\alpha = 0.9$. Particularly, we sample $4,000$ points from $\mathcal{U}([0, 50])$ as the training data. For computing the energy cost, we sample 20 trajectories initiating from the initial value $x(0) = 20$, using the linear control and the neural stochastic control of the AS along the time interval $[0, 1]$. The random seeds are set as $\{4i + 1, i = 0, \cdots, 19\}$. We compute numerically the stopping time $\tau_\epsilon$, as defined in Theorem 4.1 and with $\epsilon = 0.1$, for each trajectory, and further compute the integration of $\int_0^{\tau_\epsilon \wedge 1}[u(x(t))]^2 \mathrm{d}t$ along each trajectory. Finally, we set the average of these integrations as an estimation of the energy cost for the respective control processes. We further conduct comprehensive experiments for many different values of $k$ and compare their numerical performance to the proposed AS control method. The results are shown in Figure 15. We can see that AS control significantly outperforms linear control in terms of energy cost and convergence stability.

### A.5.6 Data-Driven Pinning Control for Cell Fate Dynamics

Indeed, our frameworks can be extended to the model-free version via a combination with existing data reconstruction method. It should be noted that the proposed methods only depend on the samples of vector field in the learning stage instead of the explicit formula of systems (see Algorithm 1,2), implying that our frameworks inherently include the model-free situation. To

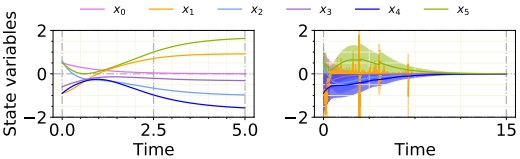

Figure 17: Pinning control for cell fate dynamics.

be concrete, we show that our framework can combine with Neural ODEs (NODEs) (Chen et al., 2018) to learn the control policy from time series data for the Cell Fate system (Sun et al., 2017; Laslo et al., 2006), which describes the interaction between two suppressors during cellular differentiation for neutrophil and macrophage cell fate choices. The system $\dot{\boldsymbol{x}} = f(\boldsymbol{x})$, $\boldsymbol{x} = (x_1, ..., x_6)$ has three steady states: $P_{1,2,3}$, where $P_{2,3}$ correspond to different cell fates and are stable and $P_1$ represents a critical expression level connecting the two fates and is unstable. The network structure of this 6-D system is a treemap, where one root node $x_1$ can stabilize itself under the original dynamic. Hence, we choose root node $x_2$ with maximum our degree and add pinning control on it to stabilize the system to unstable state $P_1$. The original trajectory that converges to $P_2$ (left) and the controlled

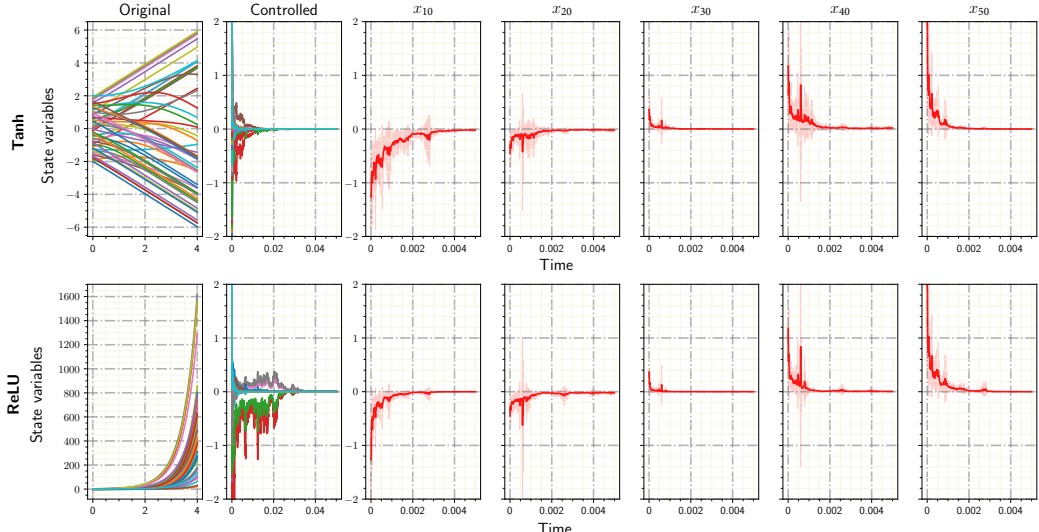

Figure 16: The sampled trajectories of the ESN (22) without and with the neural stochastic control of the AS. The 50-D ESN (22) is component-wisely displayed along the time interval $[0, 4]$ (the most left column) and the controlled ESN is component-wisely displayed along $[0, 0.05]$ (the second column from the left). The remaining panels depict the controlled dynamics of $x_{10,20,\cdots,50}$, a part of the components. Here, the solid line represents the average of the 10 realizations and the shaded area corresponds to the region which the trajectories in all the realizations sweep through. The initial values are set as $x_i^0 = -2 + \frac{i-1}{49}$ for $i = 1, \cdots, 50$.

trajectory that converges to $P_1$ (right) are shown in Figure 17. The original trajectory is used to train the NODE to reconstruct the vector field $\hat{f}$, then we use the sample of $\hat{f}$ as training data to learn our stochastic pinning control. We provide experimental details in Appendix A.5.6. The results in this section do not conflict with those in the Appendix A.9 because the 3-link pendulum does not has the similar treemap structure and self-stabilizing nodes as the cell fate system.

The stem cell fate dynamic is written as:

$$\dot{x_1} = 0.5 - ax_1,$$
$$\dot{x_2} = 5x_1/((1 + x_1)(1 + x_3^4)) - bx_2,$$
$$\dot{x_3} = 5x_4/((1 + x_4)(1 + x_2^4)) - cx_3,$$
$$\dot{x_4} = 0.5/(1 + x_2^4) - ax_4,$$
$$\dot{x_5} = (x_1x_4/(1 + x_1x_4) + 4x_3/(1 + x_3))/(1 + x_2^4) - ax_5,$$
$$\dot{x_6} = (x_1x_4/(1 + x_1x_4) + 4x_2/(1 + x_2))/(1 + x_3^4) - ax_6.$$

We set $a = b = c = 1$, then this system has 3 equilibrium points $P_1, P_2, P_3$, where only $P_1$ is unstable. Now we use data-driven method to find pinning control to stabilize $P_1$. We use coordinate transformations first to transform $P_1$ to zero for simplicity.

The connection graph structure of this model is a treemap, where $x_1, x_2$ directly affect all the other nodes. From the dynamic, $x_1$ is exponential stable in itself, hence we can pin control the root node $x_2$ to stabilize the whole system.

For model reconstruction stage, we use NODE with $(6, 50, 6)$ NN structure with $\mathcal{F}(\cdot) = $ **Tanh** to fit time series data $\mathcal{D}$ on interval $[0, 15]$ initiated from $(0.5, -0.9, 0.6, -0.6, -0.9, 0.5)$. We use standard NODE method at `https://github.com/rtqichen/torchdiffeq/tree/master/examples`. We sample the dynamic learned by NODE along $\mathcal{D}$ as the training data for AS. We learn pinning control for this dynamic that only add control to $x_2$, thus, we focus on stabilizing the sub-model for $x_2$ and treat other variables as parameters. We construct AS as $\text{Control}(1, 6, 6, 1)$ with $\mathcal{F}(\cdot) = $ **ReLU** and $\alpha = 0.9$.

### A.6 Additional Experiments

#### A.6.1 Controlling the Echo State Network

Here, we show that our proposed frameworks of neural stochastic control can perform well in controlling high-dimensional dynamics. As such, we consider the Echo State Network (ESN), one of the pioneering reservoir computing methods (Lukoševičius & Jaeger, 2009; Jaeger & Haas, 2004) and having an appealing property of dynamical short-term memory capacity (Jaeger, 2007). The reservoir states in the ESN are formulated as (Zhu et al., 2019):

$$\dot{\boldsymbol{x}} = f_{\textbf{res}}(A\boldsymbol{x} + \boldsymbol{b}), \ \boldsymbol{x} \in \mathbb{R}^n, \tag{22}$$

where the connection weight matrix $A$ is a random matrix representing the reservoir. The ESN is asymptotically stable if the spectrum radius satisfies $\rho(A) < 1$; otherwise, it is unstable for $\rho(A) > 1$ (Jaeger, 2001).

Now, we use our stochastic neural control to stabilize the unstable equilibrium for the case where $n = 50$ and $f_{\textbf{res}} = \textbf{Tanh}$ or $\textbf{ReLU}$, and $\boldsymbol{b} = \boldsymbol{0}$. Particularly, the AS is constructed as $\text{Control}(50, 200, 50)$ with $\mathcal{F}(\cdot) = \textbf{ReLU}$ and $\alpha = 0.8$. We train the parameters using the $5,000$ data sampled from $\mathcal{U}([-10, 10]^{50})$. Also we sample 10 trajectories along time interval $[0, 0.2]$ using the correspondingly trained neural stochastic control, respectively. The controlling results are shown in Figure 16.

#### A.6.2 Controlling the Lorenz System

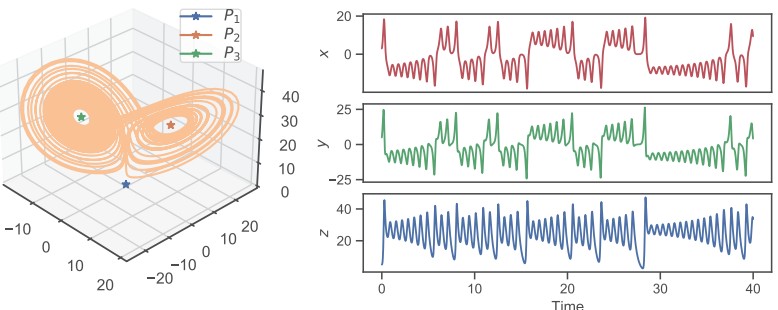

Figure 18: The chaotic dynamics of the Lorenz system (23), initiating from $(3, 5, 6)$ on the time interval $[0, 40]$. The 3-D orbits in the phase space (left) and the trajectories component-wisely displayed (right).

Here, we consider the chaotic Lorenz system (Sparrow, 2012):

$$\begin{aligned} \dot{x} &= 10(y - x), \quad \dot{y} = x(28 - z) - y, \\ \dot{z} &= xy - \frac{8}{3}z, \end{aligned} \tag{23}$$

which has three unstable equilibriums: $P_1 = (0, 0, 0)$, $P_2 = (6\sqrt{2}, 6\sqrt{2}, 27)$ and $P_3 = (-6\sqrt{2}, -6\sqrt{2}, 27)$, as shown in Figure 18. Controlling chaos is always validated by using this paradigmatic system and its controlled system (Yang et al., 1997, 2002; Ma et al., 2015). It is noted that the equilibriums $P_{2,3}$ are of the same type. So, a successful stabilization of $P_2$ implies a successful stabilization of $P_3$. Thus, in what follows, we stabilize $P_{1,2}$ using the neural stochastic control of the ES in different configurations. Precisely, we apply, respectively,

1. ES(+ICNN): $\varepsilon = 0.001$, $\text{ICNN}(12, 12, 12, 1)$, and $\text{Control}(3, 10, 10, 3)$ with $\mathcal{F}(\cdot) = \textbf{ReLU}$ and $b = 2.1$, and

2. ES(+Quadratic): $\varepsilon = 0.001$, $\text{Quadratic}(3, 12, 12, 3)$, and $\text{Control}(3, 10, 10, 3)$ with $\mathcal{F}(\cdot) = \textbf{ReLU}$ and $b = 2.1$.

For stabilizing $P_1$, we train these NNs for 200 epochs with the batch size 100 and the iterations $2,000$, using the $10,000$ data sampled from $\mathcal{U}([0, 10]^3)$ along the vector field produced by system

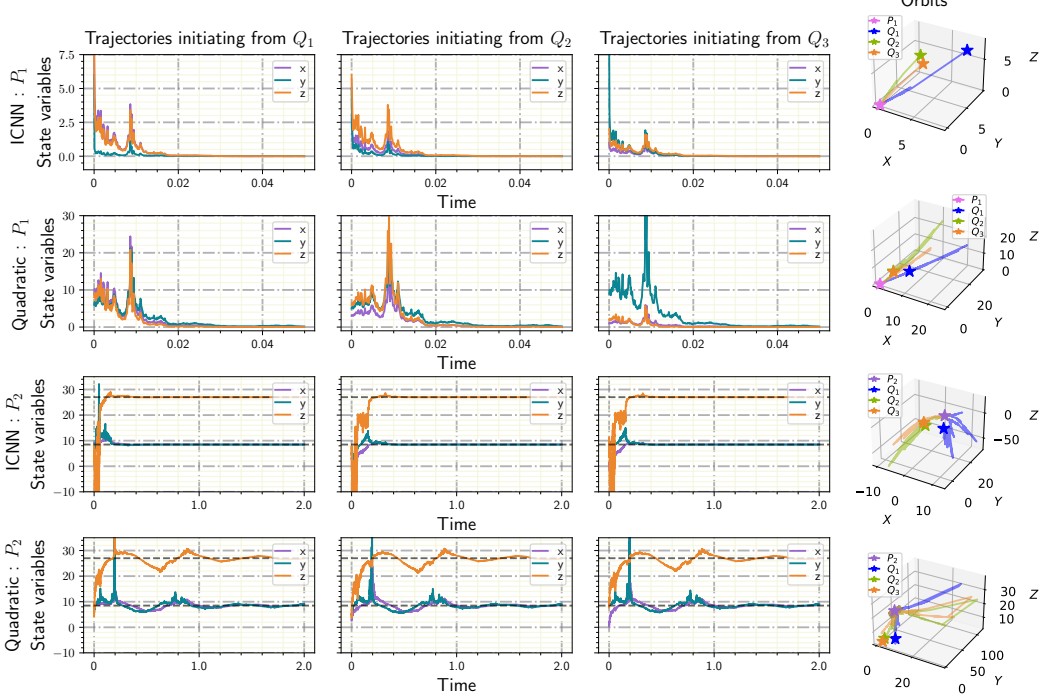

Figure 19: Controlling the Lorenz system (23) by using the neural stochastic control of the ES(+ICNN) and the ES(+Quadratic), respectively. The system, initiating from either one of the three positions $Q_{1,2,3}$, is stabilized to $P_1$ or $P_2$.

(23). For stabilizing $P_2$, we first use $\tilde{x} = x - 6\sqrt{2}$, $\tilde{y} = y - 6\sqrt{2}$, and $\tilde{z} = z - 27$ to transform $P_2$ to the zero solution of a new system, denoted by $(\tilde{x}, \tilde{y}, \tilde{z})$. Then, we train the NNs for controlling $(\tilde{x}, \tilde{y}, \tilde{z})$ in the same manner as the training procedure for stabilizing $P_1$. As for the initial values, we select as $Q_1 = (9, 6, 8)$, $Q_2 = (3, 5, 6)$, and $Q_3 = (1, 9, 2)$. The controlling results using different neural stochastic controllers are shown in Figure 19 and Table 2, respectively.

Table 2: The results, using the ES(+ICNN) and the ES(+Quadratic), are obtained through averaging the corresponding quantities produced by 20 randomly-sampled trajectories. The quantities are defined in the same manner as those in Table 1, where $T = 20$ and the thresholds for computing Ct are set as $1e{-}10$ (for $P_1$) and $0.02$ (for $P_2$), respectively.

|  | Tt | Di | Ct |
|---|---|---|---|
| ES(+ICNN): $P_1$ | 283.251s | **2.43e-45** | **1.776** |
| ES(+Quadratic): $P_1$ | **124.623**s | 4.77e-45 | 4.082 |
| ES(+ICNN): $P_2$ | 190.464s | **1.16e-5** | **2.600** |
| ES(+Quadratic): $P_2$ | **44.196**s | 0.017 | 14.823 |

## A.7 Combination with Deterministic Control

In this section, we consider different control combinations for the controlled dynamics $d\boldsymbol{x} = [f(\boldsymbol{x}) + \boldsymbol{u}_f(\boldsymbol{x})]dt + [g(\boldsymbol{x}) + \boldsymbol{u}_g(\boldsymbol{x})]dB_t$, that is, the deterministic control $\boldsymbol{u} = (\boldsymbol{u}_f, \boldsymbol{0})$, the mixed control $\boldsymbol{u} = (\boldsymbol{u}_f, \boldsymbol{u}_g)$, the proposed stochastic control $\boldsymbol{u} = (\boldsymbol{0}, \boldsymbol{u}_g)$. Similar to the proposed neural stochastic control, we can find the neural deterministic control and mixed control with the same loss functions in ES and AS. the results are shown in Figure 20. We can see that both mixed control and stochastic control outperform deterministic control in terms of convergence rate and energy cost, this illustrates the benefits of introducing the stochastic term in control. The mixed control may be more efficient because it uses both stochastic and deterministic control simultaneously. By contrast, our method allows using the stochastic term only to realize the stochastic stability while all the traditional

methods merely use the deterministic control for stabilizing the stochastic systems. This difference indicates that noise is regarded a positive factor in our stabilization, contrary to the common sense. To the best of our knowledge, there are no existing methods integrating the pure noise control with the neural networks. Therefore, our work indeed extends the existing methods non-trivially, which can be seen as a solid step for treating noise as a beneficial part with provable stability guarantee using neural networks.

Here, for all three kinds of neural controllers, we use AS Control(2,6,6,2) with $\alpha = 0.8$ and $\mathcal{F}(\cdot) = \textbf{ReLU}$, we train the NNs using the $1,000$ data sampled from $\mathcal{U}([-10, 10]^2)$. We sample $10$ trajectories along time interval $[0, 0.5]$ using the correspondingly trained neural controllers, respectively.

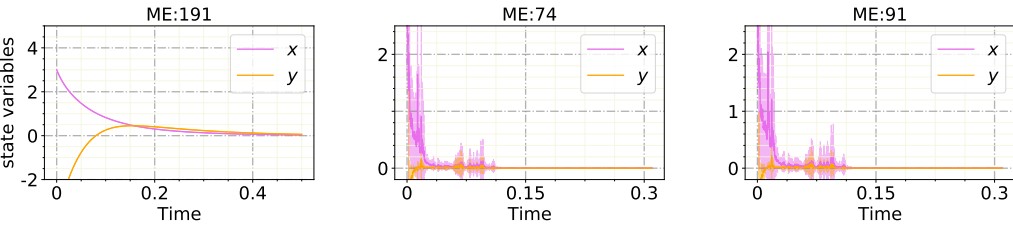

Figure 20: Numerical performance of different control combinations in the inverted pendulum experiment: deterministic control(Left), mixed control(Middle), stochastic control(Right), the mean energy for control process $\mathcal{E}(\tau_{0.1}, 0.5)$ is computed from 10 random seeds $\{2i + 1, i = 0, \cdots, 9\}$.

## A.8   Roles of Hyperparameters

Here, we investigate the role of the hyperparameter $b$ in using the ES and the role of the hyperparameter $\alpha$ in using the AS. The investigations are performed on the physical example, the inverted pendulum system (16).

● **The role of** $b$.   We test the performances of the ES(+Quadratic) in the stabilization of system (16) for different values of $b$. Here, the values of $b$ are equally spaced in $[1.0, 3.0]$. We configure the ES(+Qadratic) as Quadratic$(2, 6, 2)$ with $\mathcal{F}(\cdot) = \textbf{Tanh}$ and Control$(2, 6, 6, 2)$ with $\mathcal{F}(\cdot) = \textbf{ReLU}$. We train the NNs using $500$ data sampled from $\mathcal{U}([-10, 10])$. Furthermore, for each $b$ and the corresponding ES(+Qadratic), we sample $5$ controlled trajectories along the time interval $[0, 2]$.

In Figure 21, we show the average convergence position of the variable $\theta(t)$ over the $5$ sampled trajectories for a given $b$. The closer the position to zero, the better the neural stochastic controller performs. Clearly, the controller does not perform very well as $b$ is either large or small. Actually, $b$ can be regarded as an upper bound for the exponential decay rate of the solution. A smaller $b$ corresponds to a lower convergence rate, while a larger $b$, theoretically, can speed up the convergence. However, in applications, it is hard to treat the case of the larger $b$ because it requires more complex structures of the NNs and more training data for finding the effective Lyapunov function and the useful stochastic control function.

Through the simulations, we select five optimal values, $\{1.9, 2.0, 2.1, 2.2, 2.3\}$, for $b$ and compare their performances in training process and control process in details. The corresponding results are displayed in Figure 22. Clearly, the best value for $b$ is $2.1$, and, at this value, the correspondingly-constructed Lyapunov function $V$ and the control function $u$ have a property of the stronger convexity.

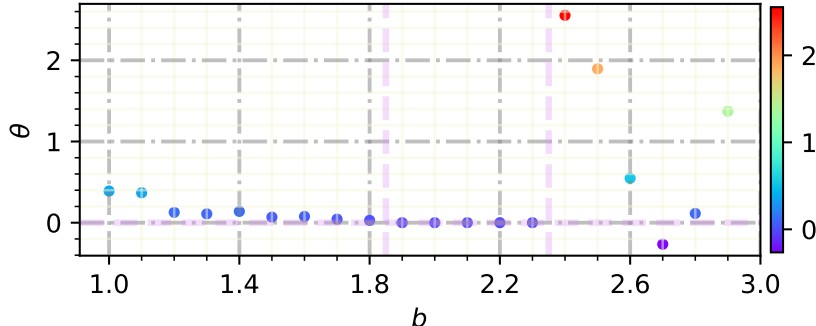

Figure 21: The convergence positions of $\theta(t)$ in the controlled system (16) for different values of $b$ selected from $\{1.0, 1.1, \cdots, 2.9\}$. Here in the simulations, the time for the convergence position is set at $t = 2$, and the position for each $b$ is obtained through averaging the quantities of the 5 sampled trajectories.

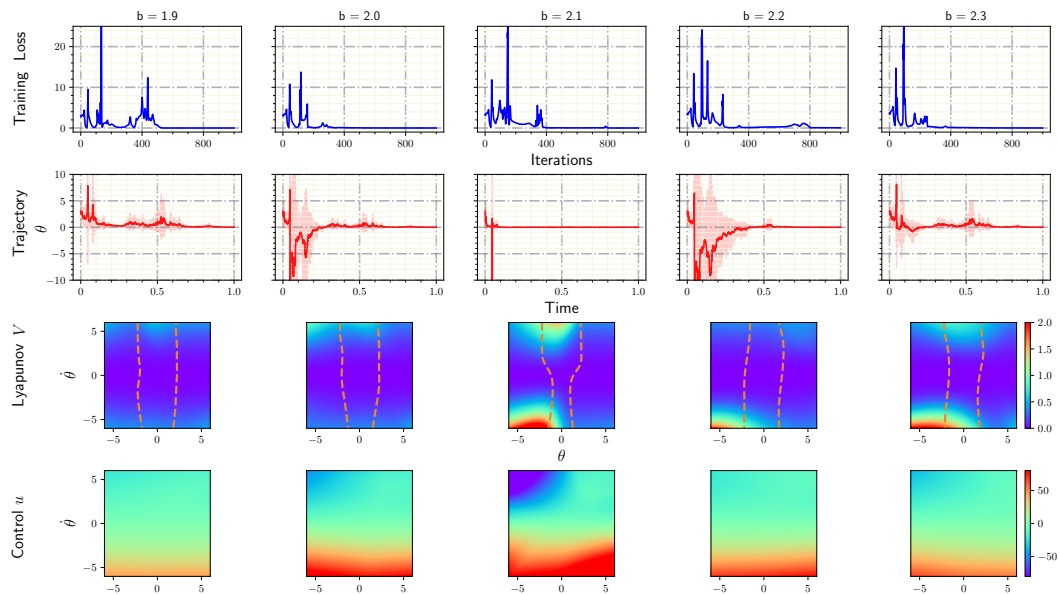

Figure 22: Training and control results of the controlled system (16) for the hyperparameter $b$ in the ES taking values from $\in \{1.9, 2.0, 2.1, 2.2, 2.3\}$, respectively. Here, depicted are the training losses (top row), the averages of the variable $\theta$ on the 5 sample trajectories initiating from $(3, 5)$ along the time interval (second row), the constructed Lyapunov functions $V$ (third row), and the constructed control functions $u$ (bottom row). The dashed lines in the panels in the third row correspond to the contour lines of $\{V \leq 0.05\}$.

● **The role of $\alpha$.** Next, we test the performances of the AS in the stabilization of system (16) for different values of $\alpha$, where the values of $\alpha$ are equally spaced in $[0, 1]$. To this end, we construct the AS as Control$(2, 6, 6, 2)$ with $\mathcal{F}(\cdot) = \textbf{ReLU}$. We sample 500 points from $\mathcal{U}([-10, 10])$ as the training data. For each $\alpha$ and the corresponding AS, we sample 10 controlled trajectories along the time interval $[0, 0.6]$. We depict the average final position of the variable $\theta(t)$ over the 10 sampled trajectories in Figure 23. Clearly, the control efficacy becomes better and better with an increase of $\alpha$.

Moreover, through the simulations, we select seven optimal values, $\{0.65, 0.7, 0.75, 0.8, 0.85, 0.9, 0.95\}$, for $\alpha$. As clearly shown in Figure 24, the convexity of the correspondingly-constructed control function $u$ becomes stronger as the value of $\alpha$ increases.

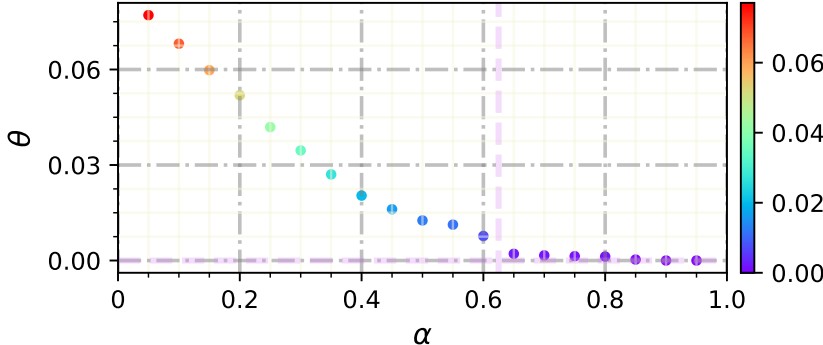

Figure 23: The convergence positions of $\theta(t)$ in the controlled system (16) for different values of $\alpha$ selected from $\{0.05, 0.1, 0.15, \cdots, 0.95\}$. Here in the simulations, the time for the convergence position is set at $t = 0.6$, and the convergence position for each $\alpha$ is obtained through averaging the quantities of the 10 sampled trajectories.

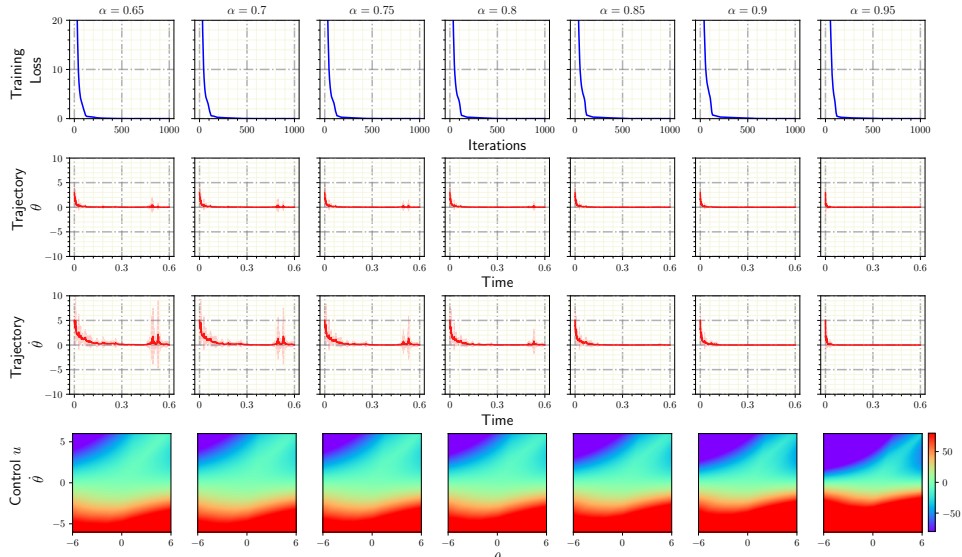

Figure 24: Training and control results of the controlled system (16) for the hyperparameter $\alpha$ in the AS taking values from $\{0.65, 0.7, 0.75, 0.8, 0.85, 0.9, 0.95\}$, respectively. Here, depicted are the training losses (top row), the averages of the variables $\theta$ and $\dot{\theta}$ on the 10 sample trajectories initiating from $(3, 5)$ along the time interval (two middle rows), and the correspondingly-constructed control functions $u$ (bottom row).

### A.9 Limitations

Here, we point out some limitations of the proposed frameworks. As we stabilize an $n$-D system with its state as $\boldsymbol{x} = (x_1, \cdots, x_n)$, the proposed controller requires each component $x_j$ $(j = 1, \cdots, n)$ to be accessible for control. In fact, our Algorithms 1 and 2 sometimes could be successfully in stabilization when taken into account is the complete control but the partial control (i.e., $x_{i_1}, \cdots, x_{i_k}$ with $1 \le i_1 < \cdots < i_k < n$ or $i_k = 1$, some components of the state, are accessible for control). However, we cannot guarantee the descent of the loss because of a lack of the stochastic stability theory for the partial control.

To illustrate it, we consider the 3-link planar pendulum. The 3-link pendulum system possesses 6 state variables $(\theta_1, \theta_2, \theta_3, \dot{\theta}_1, \dot{\theta}_2, \dot{\theta}_3)$, which represent the 3 link angles and the 3 angle velocities. Suppose that, for $i = 1, 2, 3$, each link has mass $m_i$, length $l_i$, moments of inertia $i_i$, and relative position $l_{ci}$ of the center of gravity. Thus, the controlled dynamics becomes:

$$\mathrm{d}\boldsymbol{x} = \boldsymbol{y}\mathrm{d}t + \boldsymbol{u}_1\mathrm{d}B_1(t),$$
$$\mathrm{d}\boldsymbol{y} = M(\boldsymbol{x})^{-1}[-N(\boldsymbol{x}, \boldsymbol{y})\boldsymbol{y} - Q(\boldsymbol{x})]\mathrm{d}t + M(\boldsymbol{x})^{-1}\boldsymbol{u}_2\mathrm{d}B_2(t),$$

where $\boldsymbol{x} = (\theta_1, \theta_2, \theta_3)^\top$, $\boldsymbol{y} = (\dot{\theta}_1, \dot{\theta}_2, \dot{\theta}_3)^\top$, $M, N \in \mathbb{R}^{3 \times 3}$, $Q \in \mathbb{R}^3$ with

$$M_{ij} = a_{ij}\cos(x_j - x_i), \ N_{ij} = -a_{ij}y_j\sin(x_j - x_i), \ Q_i = -b_i\sin(x_i),$$
$$a_{ii} = I_i + m_i l_{ci}^2 + l_i^2\sum_{k=i+1}^{3} m_k, \ a_{ij} = a_{ji} = m_j l_i l_{cj} + l_i l_j\sum_{k=j+1}^{3} m_k,$$
$$b_i = m_i l_{ci} + l_i\sum_{k=i+1}^{3} m_k.$$

Here, we, respectively, investigate the complete control (case 1) $\boldsymbol{u} = (\boldsymbol{u}_1, \boldsymbol{u}_2)$ and the partial control (case 2) $\boldsymbol{u} = (\boldsymbol{0}, \boldsymbol{u}_2)$ with AS control. We construct the AS as Control$(6, 24, 24, 6)$ and Control$(6, 24, 24, 3)$ with $\mathcal{F}(\cdot) = \textbf{ReLU}$. We sample 1000 points from $\mathcal{U}([-6, 6]^6)$ as the training data. We present the results in Figure 25. We can see that the complete control performs well in stabilizing 3-link pendulum systems, as guaranteed by the stochastic stability theory; nevertheless, the loss using the partial control cannot converge to zero. Generally, the partial control is more practical for utilization. Therefore, the stochastic theory assuring the efficacy of the partial control is urgently needed.

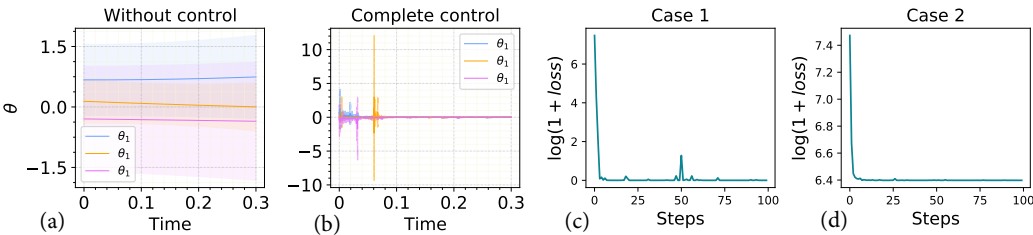

Figure 25: Dynamics of the state variables in the uncontrolled system (a) and in the controlled system using the complete control (b), initiating from $(\theta_1, \theta_2, \theta_3, \dot{\theta}_1, \dot{\theta}_2, \dot{\theta}_3)|_{t=0} \sim \mathcal{N}(0, I_{6 \times 6})$. Logarithm of the training loss in case 1 using the complete control (c) and in case 2 using the partial control (d).

Furthermore, the neural auxiliary functions can easily satisfy the soft constraint/constraints in Theorems 2.2, 2.3 due to the flexible structures of NNs and large amounts of adjustable parameters. However, we cannot guarantee the sufficient conditions are totally satisfied in the whole space because we have not solved the hard constraints optimization directly as opposed to the other existing QP or SOS based methods (Sarkar et al., 2020; Fan et al., 2020; Tan & Packard, 2004). Although we pay strictness for flexibility, our neural controller can be efficient in the control process due to the trained feedback form in advance, while the other methods solve the hard constrained optimization for each state at each time step in the control process. Hence, we can trade off between practicability and strictness according to the specific problems.

### A.10 More Details on Related Work

**Benefits of Stochastic Component** The stochastic stability theory have been systematically and fruitfully achieved in the past several decades Mao (1991, 1994a). The stochastic stability criteria often use the Lyapunov-like conditions Mao (2007), which even includes the degenerated case of the stability in ODEs. Most of the established criteria regard the noise as a negative effect impacting the stability; however, there are still some advances on the positive effect of stochasticity in the stabilization of the dynamical systems. For example, the environmental Brownian noise can suppress the explosion of the population dynamics Mao et al. (2002), a certain amount of noise can suppress the exponential growth of SDEs to the polynomial growth or more general growth form Deng et al. (2008); Caraballo et al. (2003), and the multiplicative noise can stabilize and destabilize nonlinear or hybrid differential equations (Appleby et al., 2008; Mao et al., 2007). These, therefore, motivate us to

develop *only* neural stochastic control to stabilize different sorts of dynamical systems in this article. Moreover, the stochastic stability theory has been further generalized to the other systems, such as the stochastic difference equations Appleby et al. (2006), the stochastic functional differential equations Appleby (2003), the linear PDEs with Stratonovich's noise Caraballo & Robinson (2004), and the Markov switched stochastic differential equations Wang & Zhu (2017). All these generalizations could provide the directions for further investigations on neural stochastic control policies.

**Robustness of NODE**    Inspired by the residual neural networks (He et al., 2016), the neural ODE (Chen et al., 2018) method was established to model the continuous-time dynamical system by approximating the original dynamics of this system with trainable layers. Recently, many researchers have conducted studies on robustness and adaptability of neural ODEs. In (Yan et al., 2019), the authors explored the intrinsic robustness of neural ODEs via the translation invariance of time-invariant system, and propose TisODE to improve the robustness of deep networks. In (Xie et al., 2019), the authors designed a special filter with feature denoising property that can remove the perturbation's pattern from feature maps, which can also be combined the NODE method to improve the robustness.

**Generate CLFs**    In this paper, we use neural networks to find the auxiliary functions with soft stability guarantee, while the existing work just use the hard constrained optimization to generate the auxiliary functions, i.e. CLFs. In (Tan & Packard, 2004), the authors propose to use the sum of squares optimization to generate a polynomial CLF for deterministic controls. This process is not using gradient descent with a soft loss function but is rather solving a constrained optimization (SDP) to generate the CLF itself. In (Parrilo, 2000), the method of polynomial approximations for the dynamics and the search of SOS as Lyapunov functions through SDP is given systematically. Further, (Leong et al., 2016) provides some methods for generating a CLF for stochastic systems using the Hamilton-Jacobi-Bellman formulation.