# OpenReview forum: "Neural Stochastic Control"
_NeurIPS.cc/2022/Conference — NeurIPS 2022 Accept_

### Official Review · Reviewer_Eycg · 2022-06-27

**Rating:** 7
**Confidence:** 3
**Soundness:** 3 good
**Presentation:** 3 good
**Contribution:** 4 excellent

**Summary:**

Prior work on approximating a Lyapunov-like function as a neural network focused on deterministic stabilization where the controls affect the system deterministically. For a system, for which the stochastic noise process can have a non-linear impact on the system, the gains on the noise in itself can be controlled to stabilize or de-stabilize the system. Generating such gain controls requires developing a Lyapunov-like function that satisfies certain criteria. This paper extends previous work that models such functions as a neural network. The hard constraints are converted into a soft loss function and minimized using gradient descent on the neural network. The novelty is in extending deep Lyapunov functions to stochastic stabilization where the noise process itself can be fed back with suitable gains to stabilize the system.

**Questions:**

For the harmonic linear oscillator, how would an optimal deterministic controller (u=K_{lqr}x) perform compared to the stochastic counterparts. Maybe, previous work showed that neural lyapunov controllers perform better than LQR.

It would be better if the authors can present the preliminaries in section 2 more elaborately to enable deeper understanding.

Is the derivative operator an extension of the Lie-derivative operator? The derivative operator here is related to Ito’s lemma from stochastic DDE literature.

Similar to SOS (sum of squares) methods, are there methods that don’t use deep learning to generate V. Is it possible to compare to these? Also, [A] uses deep RL for stochastic control. How does your work qualitatively compare to that?

Please try to make the figures more readable.

[A] Deep neural networks algorithms for stochastic control problems on finite horizon: numerical applications - Bachouch  et al


**Limitations:**

I don't see any potential negative social impact.

The general technique of learning lyapunov-like functions is limited to lower-dimensional systems as the methods require sampling to enforce the constraints.

**Strengths And Weaknesses:**

The experiments on Stuart-Landau equations, Lorenz oscillator are new contributions and provide strong evidence for the paper’s hypotheses. Instead of convergence to an equilibrium point, the controller is optimized to ensure convergence to a manifold of constant phase for synchronization. This can have real-world applications.


Lyapunov stability theorems have been studied in non-linear dynamical systems for a long period of time. This paper uses existing theory to bring new problems to the fore. Original theoretical contributions are interesting but not as strong a contribution as they are applicable to the linear controller only.

---

> ### Author Response · Authors · 2022-08-02
> **Thank you for your comments**
>
> We would like to thank the reviewer for the overall positive feedback and helpful suggestions. We revised the paper carefully according to the reviewer’s comments. Our major changes in the revised paper are listed in **“Response to all reviewers”**.
> ```
> Q1: For the harmonic linear oscillator, how would an optimal deterministic controller (u=K_{lqr}x) perform compared to the stochastic counterparts?
> ```
> **Response**: Thanks for your careful reading and helpful suggestion. We compare the performance of our learning control with existing methods in Figure 6, it can be seen that our method outperforms the LQR method.
> ```
> Q2: Is the derivative operator an extension of the Lie-derivative operator? The derivative operator here is related to Ito’s lemma from stochastic DDE literature.
> ```
> **Response**: Thanks for your comment. Our derivative operator is derived from Ito's formula and it can be understood as the Lie-derivative operator in the stochastic version, but in the existing literature, mathematicians do not use 'Lie-derivative' to denote this operator.  Actually, compared to the tranditional Lie-derivative, additonal terms, induced by the stochastic configuration, are included in this operator.
> ```
> Q3: Similar to SOS (sum of squares) methods, are there methods that don’t use deep learning to generate V. Is it possible to compare to these?
> ```
> **Response**:  Many thanks for your comment. There exist some quadratic program (QP) based methods that utilize the V function in SDE to dynamically find the control, but they always fix some candidate V functions and focus on generating the control. Here we provide a numerical comparison with these methods, including HDSCLF and BALSA, in Figure 6. And we can see that our learning control outperforms these methods in those experiments.
> ```
> Q4: Also, [A] uses deep RL for stochastic control. How does your work qualitatively compare to that?
> ```
> **Response**: Many thanks for your comments. Our work mainly focuses on using stochastic control to stabilize the dynamics to the target state. In other words, we pay more attention to the stabilization problem instead of the optimal control problem.   Bachouch et al mainly consider optimal control problems in the discrete-time MDP with noise over a finite horizon, they aim to minimize the cost in the finite control process. Hence, we study different problems, and we learn control function from the drift and diffusion terms while in [A] they find the optimal control dynamically.
>
> [A] Deep neural networks algorithms for stochastic control problems on finite horizon: numerical applications - Bachouch et al
>
> We would like to thank the reviewer again for your valuable comments and thoughtful suggestions. We thereby improve the paper not only from the theoretical aspect but also from the experimental aspect. We hope that the revised paper has been much improved. We do hope that the response and the revised paper, especially the added experiments with the corresponding descriptions, have addressed the main concerns fully. We would appreciate it very much if you could support our work and we are willing to make further improvements with any constructive feedback from your side.

---

> > ### Comment · Reviewer_Eycg · 2022-08-05
> > **Some additional comments**
> >
> > Thank you for the clarifications,
> >
> > A general issue with neural Lyapunov functions is that loss function can only satisfy a soft constraint/constraints as opposed to solving an hard constrained optimization. This is the price one pays for flexibility and can be discussed as a limitation of the approach.
> >
> > Further, a general issue of cancelling terms is that if the cancellation is not perfect, the residuals will excite the deterministic system and this can cause bad behavior in certain cases. While extending to the model-free case, the learned model is not going to be perfect as well. The robustness and adaptability of the policy to model errors is worth exploring.
> >
> > The additional theorems seem to be adding value. However, I have not checked the proofs thoroughly.
> >
> > The additional experimental comparisons [Fan2020, Sarkar2020] are promising. In the comparing papers, the stochastic CLF is completely provided beforehand or is a quadratic and can be highly suboptimal. This will limit the optimality of the QP policy as well. For example, how about using $V(x)=x^{T}Px$ with $P$ from LQR.  In this paper, a neural function is used which provides a lot of flexibility and can generate better stochastic stabilizing controls. Rather, is it possible to just use the sum of squares optimization to generate a polynomial CLF which does not require minimizing a soft loss function. A simple overview of the process for deterministic controls is provided in [2]. This process is not using gradient descent with a soft loss function but is rather solving a constrained optimization (SDP) to generate the CLF itself. Further, [1] provides some methods for generating a CLF for stochastic systems using the HJB formulation.
> >
> > From a high-level, it would be useful to provide an overview of applications where stochastic control is useful and whether the stochastic component can always be observed to be used by the policy, for a general audience in a ML conference.
> >
> > [1] Leong, Yoke Peng, Matanya B. Horowitz, and Joel W. Burdick. "Linearly solvable stochastic control Lyapunov functions." SIAM Journal on Control and Optimization 54.6 (2016): 3106-3125.
> >
> >
> > [2] Tan, Weehong, and Andrew Packard. "Searching for control Lyapunov functions using sums of squares programming." sibi 1.1 (2004).
> >
> >
> > [Fan2020] Fan, D. D., Nguyen, J., Thakker, R., Alatur, N., Agha-mohammadi, A. A., & Theodorou, E. A. (2020, May). Bayesian learning-based adaptive control for safety critical systems. In 2020 IEEE international conference on robotics and automation (ICRA) (pp. 4093-4099). IEEE.
> >
> >
> > [Sarkar2020] Sarkar, M., Ghose, D., & Theodorou, E. A. (2020). High-relative degree stochastic control lyapunov and barrier functions. arXiv preprint arXiv:2004.03856.

---

> > > ### Author Response · Authors · 2022-08-05
> > > **Thank you for your additional comments**
> > >
> > > Great thanks to your careful reading and further suggestions. We revised the paper carefully according to the reviewer’s newly comments.
> > >
> > > The flexibility of neural Lyapunov functions satisfying the soft constraints and the rigidity of traditional methods to meet hard constraints do need to be discussed further, and we summarize these properties in limitations part in Appendix 9.
> > > Your supplement for drawbacks of cancelling terms is indeed convincing. The robustness and adaptability of the policy to model errors in model-free case is an interesting problem needed to be further explored in our futhre work, here we mention some current works in Appendix 10, and thanks again for your valuable advice.
> > >
> > > For the additional experimental comparison, the simple candidate $V=\frac{1}{2}x^\top Px$ with $P=I$ does  restrict the optimality of QP policy. To lift the optimality, we update $P$ as LQR solution in HDSCLF, and we update $P$ from Lyapunov equation $A^{\top}P+PA=-Q$ with positive definitely $Q$ in BALSA. And we replace the original results in Figure 6.
> > >
> > > The candidate V function in stochastic setting may be found with SOS methods, but in this paper we focus on finding the controller, which cannot be accomplished from the direct use of the existing SOS methods. We can make more efforts on finding both SCLFs and controllers with SOS in our future work. We provide overview of existing methods for generating CLFs in Appendix 10.
> > >
> > > We supplement the overview of benefits of stochasticity in stabilizing systems in Appendix 10. We will continue to complete the overview of applications where stochastic control is useful and whether the stochastic component can always be observed to be used by the policy.
> > >
> > > Finally, we would like to thank the reviewer again for his/her time and positive feedback on the paper. The revised paper is improved a lot from the theoretical and experimental aspects. We hope that the reviewer will be satisfied with the revised paper and the responses as well, and then consider revising the assessment in support of the revised paper. We may make further improvements according to your feedback.

---

### Official Review · Reviewer_a1wg · 2022-07-05

**Rating:** 5
**Confidence:** 4
**Soundness:** 3 good
**Presentation:** 3 good
**Contribution:** 2 fair

**Summary:**

This paper gives an iterative procedure for computing model-based controllers that stabilize the system with noise. In particular, the control law enters the equations in the form $u(x_t)dB_t$, where $B_t$ denotes Brownian motion. The controllers are designed based on two sufficient conditions for stochastic stability, one requiring a Lyapunov function and the other working directly on the terms of the SDE. These sufficient conditions are used to derive loss functions for optimization-based search algorithms. For the Lyapunov method, both the Lyapunov function candidate and the control law are parameterized by neural  networks. The controller is computed by optimizing the resulting loss with respect to the network parameters over a randomly chosen set of states. In the other approach, no Lyapunov function is needed, and so only the control parameters are optimized. The method is tested in a variety of examples.

**Questions:**

From what I can tell, the notation $\mathbf{u}$ is overloaded to denote simultaneously the control strategy $u(x)$ and the neural network parameters of the control strategy. Is this correct? If it is, it is a bit confusing and an alternative notation to denote the function and the parameters is advisable.

**Limitations:**

They listed this as N/A, and I agree with the assessment.

**Strengths And Weaknesses:**

Strengths:
This paper shows how controllers can be designed to solve some interesting and non-standard control design problems. Namely, controlling the system via noise is theoretically established, but substantially less studied than noise-free control. The results show promising performance for a variety of nonlinear stabilization tasks.

Edit: The newer results based on learning the dynamics and partial control show promise. I would still like better comparisons, but this merits a score increase in my view.

Weaknesses:

It is unclear if the methods in this paper have any advantages over well-known stabilization methods. So, while the methods outperform simple methods that inject noise for control, it is unclear if the method has any benefit over more standard methods that do not inject noise.

More explanation on the comments above: The methodologies are model based. Furthermore, in all examples where the dynamics, (5), are written explicitly, they are fully actuated. In other words, the inputs directly affect all states. In all cases, very simple feedback-linearization schemes can stabilize these systems. Presumably, other schemes can stabilize them as well.

First we consider the case that the only noise is actually due to the controller. This is  the case in the inverted pendulum and the Stuart-Landau systems. This is presumably also the case for the coupled Stuart-Landau oscillators, the echo state network, and the Lorenz system. However, (5) does appear to be written explicitly for these systems, so this is just a guess. Anyways, when the system is fully actuated and the model is fully known, and the system has no intrinsic noise (i.e. no term $g(x)dB$), we could alternatively write the system as
$$
\dot x = f(x) + u(x).
$$
a simple controller of form
$$
u(x) = -f(x) - x
$$
Sure, we could make the problem harder by injecting noise, but in the fully actuated case, this will do.

In the simple harmonic oscillator problem, the system has the form:
$$
dx = Ax + (g(x)+u)dB,
$$
but again it is assumed that the system is fully actuated. Furthermore, $A$ is Hurwitz.  So, inn this case, $g(x)$ could be cancelled by the controller leaving a stable system.

A more minor weakness is that the theoretical results are completely restricted to model-based stochastic control, and so the learning component is not very strong.

---

> ### Author Response · Authors · 2022-08-02
> **Thank you for your comments (1/2)**
>
> We thank you for your valuable comments and helpful suggestions on the proposed methods. We recommend that the reviewer reads the **“Response to all reviewers”**. For the minor/major points in this work, we respond to them one by one. Hopefully, the following answers could help you understand our work better.
> ```
> Q1: If the method has any benefit over more standard methods that do not inject noise?
> ```
> **Response**: Thanks for your valuable comments. We supplement three numerical experiments to demonstrate the benefit of the proposed method over traditional deterministic methods. First, we compare the performance of our proposed method with existing deterministic control methods that can stabilize the SDEs, including quadratic program based HDSCLF, BALSA, and classic LQR. The results in Figure 6 show our proposed method outperforms the other methods in the convergence rate. Second, we conduct comprehensive experiments for linear control with different strengths k=2,4,6,8,10 to compare the performance with the proposed method in Figure 14 in Appendix 5.5, it can be seen that our method outperforms the linear control in energy cost and convergence stability. Thirdly, we provide numerical experiments of different control combinations on the inverted pendulum in Appendix 7 and Figure 19. The results suggest that stochasticity in control can decrease the energy cost and accelerate the convergence process. All these experiments can demonstrate the benefits of saving energy and accelerating the control process by introducing the stochasticity in control policy.
> ```
> Q2: The methodologies are model based and the dynamics are written explicitly.
> ```
> **Response**: Thanks for your careful reading and instructive comments. Although we write all the dynamics explicitly in our experiments, our proposed framework can cover the model-free case because only required are the samples of vector field instead of the explicit formula during our learning period. Thus, we can naturally extend our framework to the cases where the only time series of the systems are known. We can first use some data reconstruction methods such as NODE [Chen2018] to fit the corresponding vector field, then we sample this surrogated model to train our NN control. We illustrate the efficacy of this extended data-driven method on a cellular differentiation network model in Appendix 5.6. Thanks again for pointing out this question which can improve the generalization of our method.
> ```
> Q3: Very simple feedback-linearization schemes can stabilize the systems with explicit known dynamics.
> ```
> **Response**: Thanks for your comments. In proposition 3.2, we give a counterexample where neither the deterministic linear control nor the stochastic linear control can globally stabilize the dynamics in Eq. (9).  Here, we focus on global stability. In fact, we can surely stabilize the system from a fixed initial point with large enough linear control.  However, once we fix the linear control and change the initial point, we cannot guarantee the stability behavior of the trajectory. In this case, an appropriate nonlinear control should be designed elaborately to guarantee global stability, such as our newly proposed method.
> ```
> Q4: Once the model is fully known, the deterministic term f(x) and noise term g(x) could be cancelled by the controller.
> ```
> **Response**: Thanks for your comments. There are several reasons that we cannot directly cancel the original term. Firstly, we have shown through our experiments that our learning control is more energy saving than existing methods, this is thanks to the relatively simple structure of our NN control, while the complexity from the original dynamics is only set as the loss function. If we add -f(x) (-g(x)) in our control to cancel the original drift (diffusion) term, we may get a larger energy cost and this is not feasible in practical application. Second, we study the synchronization problem in our experiment, and this kind of model has physical or biological background such as synchronizing the neurons in the brain where the random coupled force comes from their structure and corresponding biological action. We can only add simple and weak action to the neurons, but the strength g(x) may be large due to the interaction of millions of neurons. Hence, we cannot directly cancel the g(x) term, and the similar situation also holds for f(x). Finally, pinning control on those accessible nodes can be used to stabilize the dynamics with special structures, in these cases, we can also design stochastic pinning control. Since we only add control signals to a fraction of all the nodes in the system, we naturally cannot eliminate the dynamics of all nodes. We further provide an experiment on controlling cell fate dynamics to demonstrate the validity of our stochastic pinning control in Appendix 5.6.

---

> > ### Author Response · Authors · 2022-08-02
> > **Thank you for your comments (2/2)**
> >
> > ```
> > Q5: The theoretical results are completely restricted to model-based stochastic control, and so the learning component is not very strong.
> > ```
> > **Response**: Thanks for your instructive comments. We have extended our framework to the model-free case with a combination of existing learning methods, such as NODE. This extension enlarges the role of the learning component.
> > ```
> > Q6: The notation u is overloaded to denote simultaneously the control strategy u(x) and the neural network parameters of the control strategy.
> > ```
> > **Response**: Thanks for your comment. We surely use the same notation u to denote the control function and the corresponding parameters of NN control. To better distinguish the control function and the NN parameters, we change the notation u in Algorithm 1,2 as $\theta_u$.
> >
> > We would like to thank the reviewer again for his/her comments. Now, the quality of the revised paper has been sufficiently improved thanks to the reviewers’ comments and suggestions. Hopefully, the response and the revised paper have addressed the reviewer’s main concerns, and then reconsider the assessment in support of the revised paper. Finally, we will further answer any questions you may have.

---

> > ### Comment · Reviewer_a1wg · 2022-08-09
> > **Response to Rebuttal.**
> >
> > The revisions have improved the paper. In particular, the integration with NODES and the pinning control example make the paper much stronger.
> >
> > I think the paper would really benefit from a further round of revision that makes these contributions more of a focus, rather than small sections in the appendix.
> >
> > The *counterexample* from Proposition 3.2 is not a counterexample to my critique. I did not discuss linear control. I discussed **feedback linearization**. That is using control to cancel nonlinearities. It is a classical methodology, and can be used on the system of Proposition 3.2, and on all examples aside from the neuron one.  Even in the neuronal example, feedback linearization could be used to make $x_2$ behave like a linear system. Without actually simulating this system myself, it is difficult to know whether that is sufficient to control this system to the desired equilibrium.
> >
> > The comments that required control "may be large" should be substantiated with evidence before I am really convinced that this is a competetive method.
> >
> > I plan to up my score a bit for the new contributions, but I would still think better comparison with classical methods is required.

---

> > > ### Author Response · Authors · 2022-08-09
> > > **Thanks for your response**
> > >
> > > Thanks for your support and constructive comments.
> > >
> > > We put some revised parts in appendix due to the page limit, and we can put them in the main text if the paper is accepted.
> > >
> > > We agree your comments about the feedback linearization, and we believe that our model-free extension can generalize our method to model-free cases where traditional feedback linearization cannot be used. Besides, the feedback linearization is not physically feasible in some cases due to the underlying physics (please see [1]). Thanks again for your further explanation.
> > >
> > > We will further compare our method with classical feedback linearization methods and provide the results if the paper is accepted.
> > >
> > > Finally, we would like to thank the reviewer again for his/her time and positive feedback on the paper. We may make further improvements according to your feedback.
> > >
> > > [1] G. Chen, "From the editor," in IEEE Circuits and Systems Magazine, vol. 9, no. 3, pp. 4-4, 54, Third Quarter 2009, doi: 10.1109/MCAS.2009.933856.

---

### Official Review · Reviewer_7JKN · 2022-07-11

**Rating:** 4
**Confidence:** 3
**Soundness:** 3 good
**Presentation:** 2 fair
**Contribution:** 2 fair

**Summary:**

This paper deals with stochastic problems and is in alignment with previous neural network studies inspired by the Lyapunov stability theory. In this paper, the stochastic controller is defined as a neural network that stabilizes the "zero" solution and is applied $dB_t$ in the SDE. To efficiently obtain these solutions, the authors propose two types of controllers for neural networks in stochastic control systems: the (1) exponential stabilizer and the (2) asymptotic stabilizer. The first one actually requires a neral Lyapunov function, so it can be quite accurate with nonlinear systems but computing the loss function takes a relatively long time. The second one does not require actual function but guarantees almost-sure convergence, and it updates fast in reality. This paper shows that the proposed model outperforms the classical stochastic linear control in both time and cost for systems.

**Questions:**

1. (about Theorem 4.1) In the theorem, only the result of a stochastic linear controller ($u(x) = kx$) is presented. Are there any theoretical results on the convergence of the nonlinear ES/AS controllers?
2. (about the AS loss) Could you explain more about the theoretical background of the AS loss function? Is the AS stabilizer "safe" to use for complex physical systems?
3. (about the ES stabilier) How much model complexity can the training of ES(+ICNN) model hold in a reasonable time?

**Limitations:**

From the authors' description, I can imagine a situation that AS loss is later discovered by the authors and applied in this work mainly for its low computational costs. Therefore, it is worth questioning whether the AS loss (or the notion of asymptotic attractiveness) is appropriately presented in this paper titled "Neural Stochastic Control." First, I think overall dedicated technical descriptions for this loss (Section 2.3) are too short. As far as I understand, the only reasonable description for AS controller is Theorem 2.3. I have a concern about whether Theorem 2.3 is tightly coupled with other theoretical results such as Theorem 2.2 or whether it is a way of approximation that can simplify the aforementioned framework. Therefore, the lack of justifications for AS loss is one limitation of this work.

Another limitation is that the ES stabilizer seems to be performant, but it might take a considerable amount of time to be trained. Therefore, I have another concern that this two-fold approach is actually applicable, or only AS is the reasonable algorithm that can successfully train a controller in a limited time.

**Strengths And Weaknesses:**

This is overall a solid paper that retains theoretical and empirical novelties for challenging control problems. The technical framework can be considered novel and general since it is dealing with stochastic problems. This paper provides some convergence analyses and experimental results on several physical systems. I think this work contributes to stochastic control studies that conventionally have used linear controllers to solve such problems. Meanwhile, I also would like to point out I have a few concerns regarding the assumptions on stabilizers and experimental results. I hope to see responses to these concerns from the authors.


### Strengths

* Major
  * This work presents trainable nonlinear controllers for stochastic control problems.
  * The paper offers two distinct stabilizers which can operate with different characteristics of convergence rates and computational costs.
  * The manuscript is clearly written; I think overall analyses are technically sound.

* Minor
  * The experiments cover various physical environments. General readers can grasp how each stabilizer works through the experiments.
  * I think the techniques shown in the exponential stabilizer show a promising direction for designing physics-informed neural networks.

### Weaknesses

* Major
  * Theoretical analyses might work in ideal situations, but they seem to not fully cover addressed claims.
  * If I understand correctly, I think controlling mechanism $\mathrm{d} B_t$ is somewhat limited. For example, instead of $\mathrm{d}x = f(x)\mathrm{d}t + [g(x) + u(x)]\mathrm{d}B_t$ why not use
 $\mathrm{d}x = [f(x) + u_1(x)]\mathrm{d}t + [g(x) + u_2(x)] \mathrm{d}B_t$ for efficient stabilization?
  * I could not find meaningful comparison studies with other approaches in the experiment section. The framework proposed in this work is expected to replace classical linear controllers. Therefore, it is very important to provide comparison studies with these controllers (e.g., the classical LQR techniques). The toy experiments in Figure 4 have such results for a simple system, but it seems the performance of $u(x) = kx$ is missing for other experiments. I think this work needs a revision since comparison experiments with previous methods should be included.
* Minor
  * I think the title "Neural Stochastic Control" does not clearly explain the methodology in this paper. Maybe this has to be more specific for representing two stabilizers (ES and AS). I believe the title has to be changed to indicate that this work is for the stochastic Lyapunov theory and the stochastic asymptotic stability theory.

---

> ### Author Response · Authors · 2022-08-02
> **Thank you for your comments (1/2)**
>
> We thank the reviewer for the overall positive feedback and the valuable comments. We recommend that the reviewer reads the **“Response to all reviewers”**. We hope the reviewer could find the efforts and the improvements we made not only from the theoretical aspect and the experimental aspect. For the individual comments, we are going to respond to them one by one.
> ```
> Q1: Are there any theoretical results on the convergence of the nonlinear ES/AS controllers?
> ```
> **Response**: Many thanks for your insightful suggestion. In the revision, we provide two new theorems about the upper bound estimation of convergence time and energy cost for ES and AS, respectively, in __Theorem 4.2__ and __Theorem 4.3__. These two theorems significantly improve our analytical results, we can further study the effect of the NN controller based on the formulation of the upper bound. We provide more analysis about these two theorems in Appendix.
> ```
> Q2: Could you explain more about the theoretical background of the AS loss function? Is the AS stabilizer "safe" to use for complex physical systems?
> ```
> **Response**: Thanks for your comments. The key formula $\|x\|^2(2\langle x,F(x)\rangle+\|G(x)\|_{\rm F}^2  )-(2-\alpha)\|{x}^{\top}G(x)\|^2\triangleq q(x)$ in AS loss is derived from $\mathrm{d}||x||^\alpha=\mathrm{d}(||x||^2)^{\alpha/2}$ using Ito's formula, a standard tool for stochastic analytics.  Then, we have $\mathrm{d}\Vert x\Vert^\alpha=\alpha/2\Vert x\Vert^{\alpha-4}q(x)\mathrm{d}t+\alpha\Vert x\Vert^{\alpha-2}\Vert x^\top G(x)\Vert\mathrm{d}B_t$. Hence, the term $q(x)\le0$ can drive the solution to zero due to the negative drift.  A specific explanation is provided in Appendix 3.5. Notice that the asymptotic stability in Theorem 2.3 stands almost surely (physically) instead of the stability only with a probability $1-\varepsilon$ for some small number $\varepsilon$. So, the AS stabilizer can be physically achieved and thus safely used for complex systems.
> ```
> Q3: How much model complexity can the training of the ES(+ICNN) model hold in a reasonable time?
> ```
> **Response**: Thanks for your comment. The computational complexity for ES is $\mathcal{O}(mn^2)$ for m data in n-D dynamics due to the computation for V's hessian matrix in $\mathcal{L}V$. We provide the specific complexity comparison between ES and AS in Appendix 3.5.
> ```
> Q4:  Whether Theorem 2.3 is tightly coupled with other theoretical results such as Theorem 2.2 or whether it is a way of approximation that can simplify the aforementioned framework.
> ```
> **Response**: Thanks for your valuable comments. As we explained in the last question, Theorem 2.3 utilize Ito's formula of $\mathrm{d}||x||^\alpha$ for some $0<\alpha<1$ and aims at steering  $||x||^\alpha$ to zero with constraint used in AS loss, while Theorem 2.2 and other theoretical results consider the case of $\mathrm{d}\log V(||x||)$ or $\mathrm{d}\log||x||$. So the results in Theorem 2.2 can be seen as negative fractional polynomial growth, and the growth rate in Theorem 2.3 is negative exponential growth. Hence, these two theorems have no direct connection and the constraints in these theorems do not cover each other.
> ```
> Q5: Why not use dx=[f(x)+u1(x)]dt+[g(x)+u2(x)]dBt for efficient stabilization?
> ```
> **Response**: Thanks for your instructive comment. We provide a numerical experiments of different control combinations on the inverted pendulum in Appendix 7 and Figure 19.  The results imply that our method can also be modified to find the deterministic control, and the introduction of stochastic control can decrease the energy cost and accelerate the stabilization process. The mixed control with both deterministic and stochastic terms is surely efficient, but here we focus on the stochastic term only to realize the stochastic stability because it is a novel perspective to regard the noise as a positive factor which is different from the existing methods.
> ```
> Q6: Are there meaningful comparison studies with other approaches in the experiment section?
> ```
> **Response**: Thanks for your valuable comments. We supplement a numerical comparison with existing methods in Figure 6 to improve the validity of our proposed method, it can be seen that our method ourperforms the existing HDSCLF, BALSA, and LQR methods.

---

> > ### Author Response · Authors · 2022-08-02
> > **Thank you for your comments (2/2)**
> >
> > ```
> > Q7: The title has to be changed to clearly explain the methodology in this paper.
> > ```
> > **Response**: Thanks for your suggestion. We modify our title as "Neural Stochastic Control: Exponential Stabilizer and Asymptotic Stabilizer" to indicate the connection between stochastic exponential stability and asymptotic stability theory.
> >
> > Finally, we would like to thank the reviewer again for his/her time and positive feedback on the paper. The revised paper is improved a lot from the theoretical and experimental aspects. We hope that the reviewer will be satisfied with the revised paper and the responses as well, and then consider revising the assessment in support of the revised paper. We may make further improvements according to your feedback.

---

> > > ### Comment · Reviewer_7JKN · 2022-08-09
> > > **Response to the authors**
> > >
> > > Thank you for your revisions and clarifications. I will take these into account when discussing the paper with the other reviewers.

---

### Official Review · Reviewer_qG7C · 2022-07-12

**Rating:** 7
**Confidence:** 3
**Soundness:** 3 good
**Presentation:** 4 excellent
**Contribution:** 3 good

**Summary:**

Update: I've checked out authors' response and other reviewers' comment, and my rating stays unchanged.

In this paper, the authors proposed two frameworks of neural stochastic control, i.e., the ES (exponential stabilizer) and the AS (asymptotic stabilizer), and presenting their advantages in the stochastic control. The convergence time and the energy cost of particular stochastic neural control were computed, and the efficacy of the proposed stochastic neural control were demonstrated in control problems arising from multiple physical systems. Compared with traditional control methods in the existing literature, one major novelty the authors claimed is that, the authors treat noise as a beneficial part and design stochastic control u_g to achieve the stabilization.

**Questions:**

1. Some of the numerical performance comparison with baseline methods could be improved.  For example,  in page 7, when numerically comparing the performance of the proposed method with traditional linear controller, it was claimed that "Without loss of generality, we fix k = 6.0,...", and only describe the numerical performance for k=6.0. I'd expect the value of k to have quite some impacts on the numerical performance (convergence speed and energy cost here) of the traditional linear controller, and I'm not quite convinced by the argument of "fixing k=6.0" is "without loss of generality", in terms of "numerical performance comparison" (instead of stability). Did you conduct comprehensive experiments for many different values of k and confirm that their numerical performance stay similar to each other, and always much worse than the proposed new neural stochastic control method here? To be more convincing for such numerical comparison and advantage claiming, it would be helpful for the authors to conduct such more comprehensive performance investigations and at least share a brief summary about the results in the paper (and share configs to make sure the summarized results/(claimed benefits) could be reproduced using the shared codes).

**Limitations:**

The authors have adequately discussed the limitations of the proposed framework in the manuscript and suggested a couple of promising future research directions for further improvements.

**Strengths And Weaknesses:**

1. The problem under study is interesting and important, and the obtained results on neural stochastic control seem to be novel and valuable, which would be good contribution to the research community in this area.

2. The paper presentation is good and clear, and I found the manuscript easy to follow. The concepts and main results are described/summarized clearly with clean proofs provided, and both the advantages and limitations of the proposed methodology are well explained.

3. The numerical experiments on multiple representative physical systems are helpful, and the advantages of the proposed method is reasonably convincingly demonstrated in these numerical examples.

4. The discussed limitations seem to be reasonable & helpful, and suggested future research directions look promising.

5. Overall speaking, in my opinion, this seems to be a solid paper that is acceptable to NIPS and could benefit the research community in this research area.

---

> ### Author Response · Authors · 2022-08-02
> **Thank you for your comments**
>
> We would like to thank the reviewer for the comments and valuable suggestions. For the main changes in the revised paper, please refer to the ''__Response to all reviewers__''.
> ```
> Q1: Some of the numerical performance comparisons with baseline methods should be improved. The numerical comparison on page 7 should conduct comprehensive experiments for many different values of k to compare the performance of the proposed method.
> ```
> **Response**: Thanks for your careful reading and constructive comments. We uniformly choose k=2,4,6,8,10 in Figure 4 and plot their control performance against our learning control in terms of convergence stability and energy cost.  We carry out the experiment with the same random seeds as Figure 4, and the new results are shown in Figure 14 in Appendix 5.5, we can see that our learning control has the least fluctuation and energy cost in the control process. The code of all the supplementary experiments is shared in the same link presented in the Introduction.
>
> We thank the reviewer again for your valuable comments. We do believe that the quality of the revised paper has been improved not only from the experimental but also the theoretical aspects thanks to the reviewers’ comments and suggestions. Hopefully, the responses and the revised paper have sufficiently addressed the main concerns and then the reviewer will reconsider the assessment in support of the revised paper. We are looking forward to your feedback to make further improvements of the paper.

---

### Author Response · Authors · 2022-08-02
**Response to all reviewers**

We would like to thank the reviewers for your time, efforts, and valuable comments and suggestions, which do help us to significantly improve the quality of this work in several directions.  Accordingly, we try our best to make substantial revisions. The revised article now contains the analytical results, the extensions of the proposed neural stochastic control framework, and the additional experiments on comparison studies.  Specifically, our main changes in the revision include:
1. We add additional two analytical results, presented in __Theorem 4.2__ and __Theorem 4.3__, deriving the rigorous theoretical estimation of the convergence time and the energy cost for NN control (ES and AS), which significantly improve our theoretical results and provide convincing support for the practical validity of the ES and AS. We provide more discussions about the ES and AS based on these theoretical results in Appendix 3.4.
2. We further provide several extensions based on the ES and AS frameworks, involving:

	**(a)**. AS(ES)+NODE: reconstructing the dynamics from time series data and extending our method to the model-free case,

	**(b)**. AS(ES)+pinning control: stabilizing a cellular differentiation network model by adding the stochastic control signal(s) to a single node.

Here, the NODE (Neural Ordinary Differential Equation) is a data reconstruction method cultivated in [Chen2018]. We illustrate these extensions in an experiment on controlling the cell fate dynamics in Appendix 5.6.

3. We further conduct several additional experiments, involving:

	**(a)**. comparison with existing control methods that can stabilize the SDEs, including HDSCLF [Sarkar2020], BALSA [Fan2020], and classic LQR controller, the results are shown in Figure 6,

	**(b)**. comprehensive experiments for different values of k (2,4,6,8,10) in linear control and compare the energy cost and convergence stability with learning control in the original experiment in Section 4, the additional results are provided in Appendix 5.5,

	**(c)**. comparison of different control combinations: deterministic control, mixed control, and proposed stochastic control. the results and analysis are provided in Appendix 7.  In this work, we pay more attention to a physical phenomenon that noise can play a more constructive role in stabilizing the underlying (stochastic) system instead of a negative role as understood in common sense.

4. We provide the computational complexity for the proposed frameworks and theoretical explanation for AS loss in Appendix 3.5.
5. We revise the title to "Neural Stochastic Control: Exponential Stabilizer and Asymptotic Stabilizer" to explain the methodology of our paper.
6. Due to the lack of space, we put Algorithms 1 & 2 in Appendix 4.
7. We revise the paper carefully and supplement proofs for the new theorems and details for the additional experiments in the Appendix, including the model parameter/structures and the training configurations.

[Chen2018] Chen, R. T., Rubanova, Y., Bettencourt, J., & Duvenaud, D. K. (2018). Neural ordinary differential equations. Advances in neural information processing systems, 31.

[Fan2020] Fan, D. D., Nguyen, J., Thakker, R., Alatur, N., Agha-mohammadi, A. A., & Theodorou, E. A. (2020, May). Bayesian learning-based adaptive control for safety critical systems. In 2020 IEEE international conference on robotics and automation (ICRA) (pp. 4093-4099). IEEE.

[Sarkar2020] Sarkar, M., Ghose, D., & Theodorou, E. A. (2020). High-relative degree stochastic control lyapunov and barrier functions. arXiv preprint arXiv:2004.03856.

Finally, we thank all the reviewers again for your insightful comments. We do believe that the revised article is much improved not only from the theoretical aspect but also from the experimental aspect.  In addition, we hope that the revised article as well as the individual responses for each reviewer adequately addresses the reviewers’ concerns. We appreciate it very much if the reviewers could find the further contributions and our efforts on this revised work.

---

### Meta-Review · Area_Chair_LR4G · 2022-08-27

**Recommendation:** Accept
**Confidence:** Certain

**Metareview:**

The paper proposes two new frameworks for stochastic neural control. The methods have both theoretical and experimental proofs.
The extensive replies and additional material/experiments managed to address most concerns of the reviewers (I am also satisfied by the replies to 7JKN). As the authors pointed out, ideally some of the new material should be incorporated in the main paper rather than in the appendix.
Feedback linearization (a1wg) is indeed a powerful technique, but as the authors point out cannot be applied (completely) in all situations. Having an additional method available is valuable either way.

**Award:**

No

---

### Decision · Program_Chairs · 2022-09-14

Accept